# Guiding Masked Representation Learning to Capture Spatio-Temporal Relationship of Electrocardiogram

**Yeongyeon Na**[*]**, Minje Park**[*]**, Yunwon Tae**[*]**, and Sunghoon Joo**[†]
VUNO Inc.
{yeongyeon.na, minje.park, yunwon.tae, sunghoon.joo}@vuno.co

## Abstract

Electrocardiograms (ECG) are widely employed as a diagnostic tool for monitoring electrical signals originating from a heart. Recent machine learning research efforts have focused on the application of screening various diseases using ECG signals. However, adapting to the application of screening disease is challenging in that labeled ECG data are limited. Achieving general representation through self-supervised learning (SSL) is a well-known approach to overcome the scarcity of labeled data; however, a naive application of SSL to ECG data, without considering the spatial-temporal relationships inherent in ECG signals, may yield suboptimal results. In this paper, we introduce ST-MEM (Spatio-Temporal Masked Electrocardiogram Modeling), designed to learn spatio-temporal features by reconstructing masked 12-lead ECG data. ST-MEM outperforms other SSL baseline methods in various experimental settings for arrhythmia classification tasks. Moreover, we demonstrate that ST-MEM is adaptable to various lead combinations. Through quantitative and qualitative analysis, we show a spatio-temporal relationship within ECG data. Our code is available at https://github.com/bakqui/ST-MEM.

## 1 Introduction

The electrocardiogram (ECG) is a non-invasive heart measurement to monitor the electrical activity over time and diagnose diseases. Several supervised learning models have been developed to detect various heart diseases through ECG (Siontis et al., 2021). However, since the types of heart disease are diverse and the experienced cardiologists who can provide labels are limited, learning the ECG representation for each application (i.e., detecting various heart diseases) is challenging. Recently, self-supervised learning (SSL) for general representation has emerged in natural language processing (Kenton & Toutanova, 2019; Brown et al., 2020) and computer vision (Chen et al., 2020; Grill et al., 2020; He et al., 2020; Caron et al., 2021) since it can be leveraged for numerous tasks, such as translation, sentence classification, image classification, and image generation. In ECG-based diagnosis, there were also similar efforts to learn general representation through SSL to overcome the limited resources and detect various heart diseases.

ECG-based representation learning through SSL is usually considered in two different learning methods: contrastive and generative learning (Jing & Tian, 2020). Contrastive learning (Sarkar & Etemad, 2020; Le et al., 2023; Gopal et al., 2021; Soltanieh et al., 2022; Kiyasseh et al., 2021; Wei et al., 2022) is a method to ensure similarity in the context before and after data augmentation. These data are usually defined by numerous augmentation methods (e.g., cropping, flipping, and shifting); however, despite a simple augmentation, the information on the semantic meaning of ECG signals can change drastically (Lan et al., 2023). Generative learning is a method of learning the representation of data by reconstructing all or part of the input, such as a simple framework for masked image modeling (Xie et al., 2022) or Masked Autoencoder (MAE) (He et al., 2022). In the case of ECG, there are variants of MAE that reconstruct input ECG signals (Zhang et al., 2022;

---

[*] Equal contribution.
[†] Corresponding author.

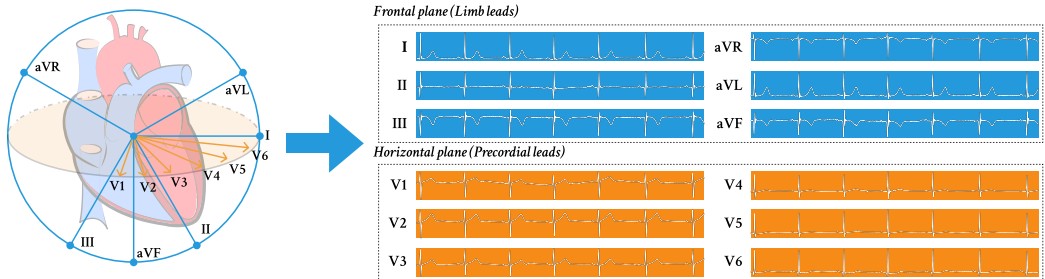

Figure 1: An illustration of 12-lead electrocardiogram (ECG). ECG signals consist of 12 leads. Each lead is measured from different spatial locations. Limb leads (i.e., I, II, III, aVR, aVL, and aVF) are generated from a frontal plane, while precordial leads (i.e., V1, V2, V3, V4, V5, and V6) are obtained from a horizontal plane.

Sawano et al., 2022; Hu et al., 2023). Generative learning is often augmentation-free or involves simpler data transformations that are more suitable for preserving the integrity of ECG data. This approach ensures that the context and meaningful information in ECG signals are retained.

In ECG-based representation learning, exploiting both spatial and temporal information in ECG is significant. For instance, if we have an $L$-lead ECG, it means that cardiac activity is obtained over a duration with $L$ views. Therefore, we can understand a heart complementary when its ECG is gained not only spatially or temporally but spatio-temporally. The most common setting is standard 12-lead ECG (i.e., a heart is observed in 12 views) as shown in Figure 1, and for some cases, ECG in which its leads are a subset of 12-lead (i.e., reduced lead ECG) is acquired.

In this work, we leverage ECG to learn general representations by introducing a simple but effective self-supervised learning framework using MAE architecture. Throughout this process, both temporal and spatial information present in the ECG is utilized.

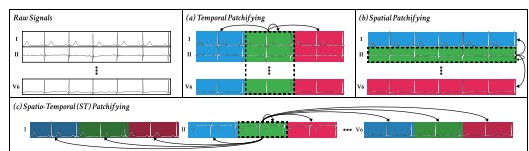

Figure 2: An illustration of spatio-temporal patchifying. The black dashed box indicates the query patch; each arrow represents the self-attention arrow; each color represents a patch, a single input sample for the model. Temporal patchifying from (a) provides three different patches (i.e., three different inputs). Spatial patchifying from (b) yields 12 patches for every 12 leads. Spatio-temporal patchifying from (c) can provide fine-grained input signals for the model, which allows for capturing spatial and temporal relationships.

The approach involves applying spatio-temporal patchifying to ECG data, as illustrated in Figure 2 (c), with lead indicators such as lead-wise shared decoder, learnable lead embeddings, and separation embedding, as depicted in Figure 3. Moreover, we show that our model can be structurally fine-tuned for reduced lead ECG, demonstrating excellent performance not only in the standard 12-lead setting but also in limb leads and single leads. Finally, through quantitative and qualitative analysis, we demonstrated that spatial and temporal features were effectively learned.

Our contributions are the following. **(1)** We propose the simple but effective ECG-specific generative self-supervised learning framework, named ST-MEM (Spatio-Temporal Masked Electrocardiogram Modeling). **(2)** ST-MEM can learn general representation by capturing spatio-temporal relationship of ECGs. We show this relationship by quantitative and qualitative analysis. **(3)** Through extensive experiments, ST-MEM demonstrates comparable performance to other contrastive and generative learning methods, which are widely used in ECG representation learning. We validate that ST-MEM excels in both fine-tuning and linear evaluation for the arrhythmia classification task. This robust performance extends across various scenarios, such as scarcity of labeled data and reduced lead settings.

## 2 BACKGROUNDS: ECG

The ECG is a non-invasive heart measurement to observe the electrical signals over time and diagnose diseases. A standard 12-lead ECG, interpreted as a multivariate time series, is the most

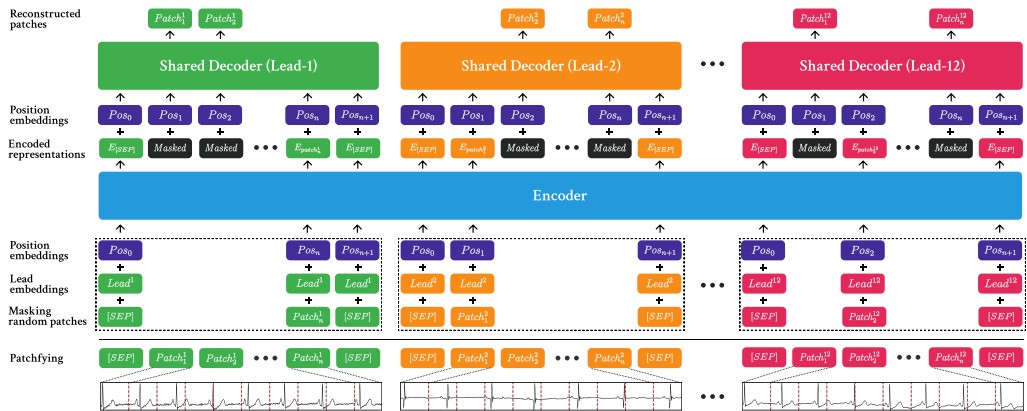

Figure 3: An overview of our proposed method. ST-MEM consists of an encoder and decoder for reconstructing the masked ECG signals. The encoder takes patchfied ECG signals with lead and position embedding. The shared decoder reconstructs the masked ECG signals for each lead by utilizing the encoded representations.

common measurement setting that provides spatial and temporal information regarding the heart. As shown in Figure 1, the 12 leads are I, II, III, aVR, aVL, aVF, V1, V2, V3, V4, V5, and V6, respectively, which are electrical signals measured at different locations of the heart.

As we mentioned above, ECG captures both spatio-temporal information. Initially, it provides temporal insights by monitoring the electrical activity of the heart continuously. This is achieved through the detection of voltage fluctuations within the cardiac muscle during each cardiac cycle, which are then represented as waveforms over time. Consequently, it displays multiple cycles of heart electrical activity spanning from one beat to the next. Moreover, ECG is not just a single measurement but rather involves multiple leads. Leads are like different views of the heart's electrical activity from different angles. In a standard 12-lead ECG, there are 12 different leads, each offering a distinct spatial perspective of cardiac electrical activity. The limb leads (I, II, III, aVR, aVL, and aVF) are placed on the arms and legs, providing frontal plane views, and the precordial leads (V1 to V6) are placed on the chest, giving anterior-posterior(or horizontal plane) views.

Although the standard 12-lead ECG remains significant, the utilization of its reduced lead sets should be considered as well. The advancement of mobile devices, such as smartwatches that are capable of ECG measurements, has led to a substantial increase in limb leads and single lead data. Obtaining reduced lead sets like limb leads or a single lead is often preferred. Therefore, in this work, we demonstrate our ECG representation not only on the 12-lead setting but also on reduced lead settings.

## 3 METHOD

### 3.1 SELF-SUPERVISED PRE-TRAINING WITH SPATIO-TEMPORAL MASKED AUTO-ENCODER

In this section, we present ST-MEM, a self-supervision framework for general representation learning for ECG. This incorporates the spatio-temporal relationship, which leads to providing enhanced representation. An overview of our proposed method is depicted in Figure 3.

**Masked auto-encoder with spatio-temporal patchifying.** We consider a pre-text task to reconstruct a randomly masked portion of the data. In particular, we adopt an auto-encoder-based reconstruction task, called MAE (He et al., 2022). It consists of a vision transformer (ViT) encoder (Dosovitskiy et al., 2020) and a decoder with additional transformer blocks.

Considering an ECG signal denoted as $\boldsymbol{X} \in \mathbb{R}^{L \times T}$ with $L$ leads and length $T$, each $l$th lead signal, represented as $\boldsymbol{X}^l$, is divided into non-overlapping patches defined as $Patch^l = \{Patch_1^l, ..., Patch_n^l\} \in \mathbb{R}^{n \times p}$. Here, $n$ is determined by $T/p$, and $p$ represents the size of each patch. We independently split each lead signal into spatio-temporal patches, as depicted in Figure 2 (c), resulting in a total of $L \times n$ patches for a single 12-lead ECG signal.

The patches undergo linear projection to form patch embeddings of dimension $D$. They are added with a positional embedding, $Pos \in \mathbb{R}^D$, to create a embedding sequence as $Z = \{Z_1, ..., Z_{Ln}\} \in \mathbb{R}^{L \times n \times D}$. This embedding sequence becomes the input for the encoder. Since self-attention operations are executed among temporal patches across different leads, the encoder explicitly learns the spatio-temporal relationship within the ECG data. In the pre-training phase, we randomly select indices of embeddings, which are denoted as a masked index set $\mathcal{M}$. After masking the corresponding embeddings, the encoder only receives unmasked embeddings, $\{Z_i\}_{i \notin \mathcal{M}}$, where the number of unmasked embeddings is determined as $Ln(1-m)$ with a masking proportion $m$. We set the masking ratio to a value in $[0, 1]$.

The encoded embeddings are fed to the decoder along with a learnable shared mask embedding, defined by $Masked \in \mathbb{R}^D$. This embedding is employed to reconstruct the patches of the $\mathcal{M}$. The training objective is to minimize the discrepancy between the original raw signal values of the masked patches, $\{Patch_i\}_{i \in \mathcal{M}} \in \mathbb{R}^{L \times (nm) \times p}$, and their corresponding reconstructions, $\{\widehat{Patch}_i\}_{i \in \mathcal{M}}$. The loss function is defined as $\mathcal{L}_{\text{SSL}} = \frac{1}{Lnm} \sum_{i \in \mathcal{M}} \left\| \widehat{Patch}_i - Patch_i \right\|$.

**Lead-wise shared decoder.** Although spatio-temporal patchifying can have benefits in ECG representation learning, it can be detrimental without careful consideration, especially within the MAE framework. One drawback of spatio-temporal patchifying is that it simplifies the reconstruction task, allowing the decoder to access unmasked embeddings from different leads with identical temporal information. This ease of reconstruction can negatively impact the training of the encoder.

To address this issue, we employ a straightforward approach by limiting the decoder to operate on only one lead. We configure the decoder to process the embedding sequence of each lead independently, ensuring that it does not make explicit use of embeddings from other leads during reconstruction. This intentional design choice introduces complexity to the task, motivating the encoders to effectively learn spatio-temporal representations. Additionally, we promote training efficiency by having patch embeddings of each lead share a single decoder.

**Lead indicating modules.** Merely increasing the difficulty of the reconstruction task may not be sufficient to enable our encoders to learn spatial representations effectively. Specifically, our encoder and decoder may face challenges in differentiating the lead from which the embedding originates. To address this, we introduce additional modules aimed at enhancing the ability to distinguish between leads effectively.

First, we add a lead-specific embedding, denoted as $Lead \in \mathbb{R}^D$, for every patch embedding. This lead embedding is a function of the originating lead of the patch embedding, ensuring that all patch embeddings originating from the same lead have the same lead embedding. Next, we insert an additional shared embedding, denoted as $[SEP] \in \mathbb{R}^D$, which is added before and after each patch embedding sequence. These modules support the model to distinguish between patch embeddings from different leads (i.e., leading the model to learn the relationships between the different leads).

## 3.2 DOWNSTREAM FINE-TUNING

After pre-training through SSL, we utilize only the encoder for downstream tasks and discard the decoder. We augment the encoder with a basic linear layer, which serves as the classifier head. Subsequently, the encoder, along with the classifier head, is fine-tuned to optimize performance on downstream data across various scenarios.

**ECG classification.** For a multi-class arrhythmia diagnosis problem with $K$ classes, our model takes the ECG input, encodes it with the encoder, and passes it through the classifier head to calculate logits. The model makes class predictions, $\hat{\boldsymbol{y}} = \{\hat{y}_1, ..., \hat{y}_K\}$, by applying the softmax function to the logits. These predictions are then compared with the actual one-hot class label $\boldsymbol{y} = \{y_1, ..., y_K\}$ to minimize the loss function defined as $\mathcal{L}_{CE} = \frac{1}{K} \sum_{c=1}^{K} (-y_c \log(\hat{y}_c))$.

**Fine-tuning in reduced lead setting.** Our model demonstrates robust adaptability, even when there is a discrepancy in the number of leads between datasets used for pre-training and fine-tuning. Firstly, the shape of the input patch fed into our model remains unaffected by the number of leads, making our model agnostic to the number of leads in the input ECG. Additionally, our model learns distinctive characteristics of different leads through the lead-indicating module. As a result, our

model is well-suited for reduced lead scenarios such as pre-training on a 12-lead dataset to learn general representations and subsequently fine-tuning on a dataset with fewer leads.

## 4 EXPERIMENTAL SETTINGS

This section provides a brief overview of the datasets utilized in our experiments, as well as the baselines against which our proposed method will be compared. More detailed information regarding the datasets, baselines, and implementation can be found in Appendix A.

### 4.1 DATASETS

There are three 12-lead ECG datasets designated for pre-training, along with three other ECG datasets intended for downstream tasks. In the context of pre-training, we leverage the combined power of all three datasets. These datasets consist of *Chapman* (Zheng et al., 2020b), *Ningbo* (Zheng et al., 2020a), and *CODE-15* (Ribeiro et al., 2021). As for the downstream datasets, we used *PTB-XL* (Wagner et al., 2020), *CPSC2018* (Liu et al., 2018), and *PhysioNet2017* (Clifford et al., 2017) datasets, where *PhysioNet2017* is a single-lead dataset (i.e., lead I dataset) and others are 12-lead ECG datasets. For consistency, we resampled all ECGs, including pre-training and downstream datasets, to 250Hz.

**Pre-training datasets.** *Chapman* dataset comprises 10,646 12-lead ECG recordings, each lasting 10 seconds with a sample rate of 500 Hz. Similarly, the *Ningbo* dataset contains 34,905 12-lead ECG recordings, also 10 seconds in duration and sampled at 500Hz. On the other hand, *CODE-15* encompasses 345,779 12-lead ECG recordings from 233,770 patients, with a sample rate of 400 Hz. Within the 345,779 ECGs, 143,328 are 10-second recordings, with the remainder being 7 seconds, and we and we exclusively utilized the 10-second ECGs. In alignment with the PhysioNet 2021 challenge (Reyna et al., 2021), we removed 399 ECGs from the *Chapman* dataset. We proceeded to merge these three datasets into a unified dataset for pre-training. This pre-training dataset comprises a total of 188,480 ECGs. Note that, during pre-training, we use all ECGs without focusing on specific labels.

**Downstream datasets.** *PTB-XL* comprises 21,837 12-lead ECG recordings collected from 18,885 patients, each lasting 10 seconds and sampled at a rate of 500 Hz. This dataset has five distinct labels, including cardiac arrhythmia and myocardial infarction. In the case of *CPSC2018*, it contains 6,877 12-lead ECG recordings, ranging from 6 seconds to 60 seconds in duration, with a sample rate of 500 Hz. This dataset is characterized by nine different labels associated with cardiac arrhythmia. Lastly, *PhysioNet2017* consists of 8,528 ECGs recorded from lead I, with durations spanning from 9 to 60 seconds and sampled at a rate of 200 Hz. This dataset has four cardiac arrhythmia categories. ECGs shorter than 10 seconds are omitted, and for the remaining records, each is cropped into a consistent length of disjointed 10 seconds. These cropped ECG segments are considered individual data points and evaluated independently.

### 4.2 BASELINES

All backbones are fixed as ViT-B (Dosovitskiy et al., 2020) with different patch projection layers corresponding to the patchifying way. We compared our pre-training method to networks initialized randomly (i.e., **Supervised**) and pre-trained models that learn representation by contrastive learning and generative learning methods. In the realm of contrastive learning, we evaluated both **MoCo v3** (Chen et al., 2021) and Contrastive Multi-segment Coding (**CMSC**) from CLOCS (Kiyasseh et al., 2021). **MoCo v3** encourages the similarity between representations of instances and their augmented counterparts, while **CMSC** divides an ECG into two temporal segments and encourages the similarity of representations for these segments.

Additionally, we examined generative learning methods, specifically Masked Time Autoencoder (**MTAE**) and Masked Lead Autoencoder (**MLAE**) from MaeFE (Zhang et al., 2022). **MTAE** constructs representations by reconstructing temporal patches as depicted in Figure 2 (a), while **MLAE** does by reconstructing spatial patches, illustrated in Figure 2 (b). These models do not have lead-indicating modules (e.g., lead-wise decoder, SEP, and lead embeddings) and differ in their approach to patchifying input ECGs. Furthermore, to enhance robustness and adaptation to reduced lead sets,

Table 1: Linear evaluation and fine-tuning results of arrhythmia and myocardial infarction (MI) classification tasks. The experiment is conducted based on 12-lead ECG data on unseen data (i.e., not used during the pre-training stage).

| Methods | PTB-XL | | | CPSC2018 | | |
|---|---|---|---|---|---|---|
| | Accuracy | F1 | AUROC | Accuracy | F1 | AUROC |
| Supervised | $0.787 \pm 0.006$ | $0.604 \pm 0.010$ | $0.905 \pm 0.004$ | $0.779 \pm 0.008$ | $0.753 \pm 0.012$ | $0.958 \pm 0.002$ |
| *Linear Evaluation* | | | | | | |
| MoCo v3 | $0.552 \pm 0.000$ | $0.142 \pm 0.000$ | $0.739 \pm 0.006$ | $0.268 \pm 0.055$ | $0.080 \pm 0.038$ | $0.712 \pm 0.054$ |
| CMSC | $0.681 \pm 0.032$ | $0.441 \pm 0.058$ | $0.797 \pm 0.038$ | $0.361 \pm 0.005$ | $0.238 \pm 0.022$ | $0.724 \pm 0.013$ |
| MTAE | $0.683 \pm 0.008$ | $0.437 \pm 0.012$ | $0.807 \pm 0.006$ | $0.486 \pm 0.012$ | $0.349 \pm 0.034$ | $0.818 \pm 0.010$ |
| MTAE+RLM | $0.687 \pm 0.006$ | $0.444 \pm 0.009$ | $0.806 \pm 0.005$ | $0.480 \pm 0.010$ | $0.342 \pm 0.022$ | $0.824 \pm 0.006$ |
| MLAE | $0.649 \pm 0.008$ | $0.382 \pm 0.020$ | $0.779 \pm 0.008$ | $0.443 \pm 0.014$ | $0.263 \pm 0.021$ | $0.794 \pm 0.016$ |
| ST-MEM (Ours) | $\mathbf{0.726 \pm 0.005}$ | $\mathbf{0.508 \pm 0.008}$ | $\mathbf{0.838 \pm 0.011}$ | $\mathbf{0.723 \pm 0.008}$ | $\mathbf{0.641 \pm 0.010}$ | $\mathbf{0.938 \pm 0.002}$ |
| *Fine-tuning* | | | | | | |
| MoCo v3 | $0.799 \pm 0.004$ | $0.644 \pm 0.010$ | $0.913 \pm 0.002$ | $0.852 \pm 0.002$ | $0.838 \pm 0.002$ | $0.967 \pm 0.003$ |
| CMSC | $0.724 \pm 0.067$ | $0.510 \pm 0.115$ | $0.877 \pm 0.003$ | $0.736 \pm 0.006$ | $0.717 \pm 0.006$ | $0.938 \pm 0.006$ |
| MTAE | $0.789 \pm 0.002$ | $0.613 \pm 0.015$ | $0.910 \pm 0.001$ | $0.793 \pm 0.004$ | $0.769 \pm 0.004$ | $0.961 \pm 0.001$ |
| MTAE+RLM | $0.793 \pm 0.002$ | $0.615 \pm 0.007$ | $0.911 \pm 0.004$ | $0.782 \pm 0.002$ | $0.756 \pm 0.003$ | $0.960 \pm 0.002$ |
| MLAE | $0.802 \pm 0.004$ | $0.625 \pm 0.009$ | $0.915 \pm 0.001$ | $0.834 \pm 0.007$ | $0.816 \pm 0.009$ | $0.973 \pm 0.002$ |
| ST-MEM (Ours) | $\mathbf{0.825 \pm 0.002}$ | $\mathbf{0.655 \pm 0.003}$ | $\mathbf{0.933 \pm 0.003}$ | $\mathbf{0.872 \pm 0.009}$ | $\mathbf{0.857 \pm 0.012}$ | $\mathbf{0.980 \pm 0.001}$ |

we introduce augmentation Random Lead Masking (RLM) (Oh et al., 2022) to MTAE, resulting in **MTAE+RLM**.

# 5 EXPERIMENTS AND RESULTS

In this section, we examine the results of our experiments, evaluating them both quantitatively and qualitatively to verify the effectiveness of ST-MEM. Additional experimental results are reported in Appendix B.

## 5.1 EXPERIMENTAL RESULTS

As shown in Table 1, we evaluate the general ECG representation obtained from each SSL method by conducting both linear evaluation and fine-tuning experiments on unseen datasets (i.e., not used during a pre-training stage). *PTB-XL* is a task of classifying both myocardial infarction (MI) and cardiac arrhythmia, while *CPSC2018* consists of only cardiac arrhythmia [1]. Our proposed method, ST-MEM, shows outperforming performance against other baseline methods. In particular, we achieve a non-trivial margin with others on accuracy, F1, and AUROC scores in the linear evaluation. For instance, the F1 score is increased in the range of 0.064 to 0.366 and 0.292 to 0.561 on both *PTB-XL* and *CPSC2018* datasets. This demonstrates that ST-MEM learns general ECG representation compared to others by explicitly considering both spatial and temporal information for ECG signals. Moreover, we present the applicability of ST-MEM by adapting to each different heart disease task. Although the baseline methods are pre-trained from large ECG datasets, their performance is similar to the supervised learning method, only used with downstream datasets, *PTB-XL* and *CPSC2018*. The CMSC pre-training method is even lower than the supervised learning method when we fine-tune the encoder. However, ST-MEM shows consistent outperforming performance on task adaptation, which is fine-tuning on downstream datasets.

## 5.2 EFFECTIVENESS OF GENERAL ECG REPRESENTATION IN A LOW-RESOURCE SETTING

The diversity of heart diseases and few experienced cardiologists prevent obtaining abundant labeled ECG data; thus, relying only on supervised learning methods is not a suitable solution for the low-resource problem. We conduct low-resource experiments to demonstrate that general ECG representation can mitigate the scarcity of labeled data. As shown in Table 2, we first randomly sampled 1% and 5% of data from each dataset. Then, we fine-tune the model for each task. Although

---

[1]Cardiac arrhythmia can be divided into different sub-classes (e.g., AV block (AVB) and premature atrial contraction (PAC)) in both *PTB-XL* and *CPSC2018* datasets.

Table 2: Experiments of low-resource settings. 1% and 5% indicate the random sampling for training and validation data; however, the test data are the same for all results. Three different sampling was done, and the results were averaged. 12-lead ECG signals are used for each result, and the score represents the AUROC scores.

| Methods | PTB-XL | | | CPSC2018 | | |
|---|---|---|---|---|---|---|
| | 1% | 5% | 100% | 1% | 5% | 100% |
| Supervised | 0.676 ± 0.011 | 0.736 ± 0.020 | 0.905 ± 0.004 | 0.600 ± 0.095 | 0.609 ± 0.111 | 0.958 ± 0.002 |
| MoCo v3 | 0.797 ± 0.006 | 0.826 ± 0.015 | 0.913 ± 0.002 | 0.791 ± 0.045 | 0.903 ± 0.019 | 0.967 ± 0.003 |
| CMSC | 0.648 ± 0.064 | 0.773 ± 0.023 | 0.877 ± 0.003 | 0.625 ± 0.013 | 0.732 ± 0.038 | 0.938 ± 0.006 |
| MTAE | 0.707 ± 0.024 | 0.713 ± 0.001 | 0.910 ± 0.001 | 0.670 ± 0.032 | 0.756 ± 0.013 | 0.961 ± 0.001 |
| MTAE+RLM | 0.730 ± 0.030 | 0.730 ± 0.003 | 0.911 ± 0.004 | 0.708 ± 0.020 | 0.726 ± 0.011 | 0.960 ± 0.002 |
| MLAE | 0.793 ± 0.007 | 0.838 ± 0.018 | 0.915 ± 0.001 | 0.860 ± 0.013 | 0.922 ± 0.007 | 0.973 ± 0.002 |
| ST-MEM (Ours) | **0.815 ± 0.012** | **0.878 ± 0.011** | **0.933 ± 0.003** | **0.897 ± 0.025** | **0.952 ± 0.004** | **0.980 ± 0.001** |

Table 3: Robustness of any lead combinations. 6-lead represents limb leads, I, II, III, aVR, aVL, aVF, while 1-lead is a single lead, I. The bold represents the significant difference ($p < 0.05$) against other baseline methods. The score indicates the AUROC score.

| Methods | PTB-XL | | | CPSC2018 | | | PhysioNet2017 |
|---|---|---|---|---|---|---|---|
| | 12-lead | 6-lead | 1-lead | 12-lead | 6-lead | 1-lead | 1-lead |
| MTAE+RLM | 0.911 ± 0.004 | 0.888 ± 0.002 | 0.795 ± 0.003 | 0.960 ± 0.002 | 0.931 ± 0.017 | 0.909 ± 0.006 | 0.857 ± 0.005 |
| MLAE | 0.915 ± 0.001 | 0.890 ± 0.001 | 0.797 ± 0.001 | 0.973 ± 0.002 | 0.959 ± 0.002 | 0.925 ± 0.001 | 0.861 ± 0.003 |
| ST-MEM (Ours) | **0.933 ± 0.003** | **0.903 ± 0.007** | **0.804 ± 0.005** | **0.980 ± 0.001** | **0.973 ± 0.002** | **0.937 ± 0.006** | **0.866 ± 0.003** |

the performance of a supervised learning method drops drastically compared to 100% of data, other SSL methods still show comparable performance, which shows the need for general ECG representation learning. However, our proposed method, ST-MEM, surpasses other representation learning methods from all 1% and 5%. Moreover, in *CPSC2018*, only with 5% of the data, we could achieve a similar performance against 100% of the data result.

## 5.3 PERFORMANCE IN REDUCED LEAD SETTINGS

12-lead ECG is a standard measurement that provides whole spatial and temporal information regarding the heart. However, measuring precordial leads (i.e., V1 to V6) requires expertise, while limb leads (i.e., I, II, III, aVR, aVL, and aVF) are easily accessible through smart devices such as a smartwatch. Therefore, we need a generally applicable ECG representation that is robust not only for 12 leads but also for any lead combinations. Simple representation learning methods, such as MoCo v3, CMSC, and MTAE, do not consider the robustness of the lead combination; thus, the model requires 12 lead signals during a fine-tuning stage. On the other hand, the random lead masking (RLM) method is one of the approaches to handle any lead combinations since the model can learn the robustness of leads by masking the random leads during the pre-training stage. Our proposed method, ST-MEM, also addresses the lead combination through spatio-temporal patchifying with explicit lead embeddings. Table 3 compares ST-MEM with RLM and MLAE methods on 1-lead (I), 6-lead (I, II, III, aVR, aVL, and aVF), and 12-lead. The results demonstrate that ST-MEM is robust to any lead combinations, even in a *PhysioNet2017* dataset.

## 5.4 ANALYSIS OF GENERAL ECG REPRESENTATION LEARNED FROM ST-MEM

Incorporating spatial information in the ECG representation is significant for ECG signals. As shown in Figure 1, the ECG signals are spatially separated by frontal and horizontal planes; the former indicates the limb leads, while the latter represents the precordial leads. To validate the spatial information for the representation learned from ST-MEM, we visualize the embedding space using t-SNE (Van der Maaten & Hinton, 2008) as depicted in Figure 4. Each circle indicates the representation of one ECG sample in a particular lead. We utilize the Gaussian mixture model (GMM) (Rasmussen, 1999) to cluster those representations learned from ST-MEM. Interestingly, the samples assigned to a blue cluster are mostly limb leads, I and II, whereas the orange cluster is formed with precordial leads, V2 and V3. This phenomenon demonstrates that the representation learned from ST-MEM incorporates the spatial information of ECG signals. In addition, in Table 4, we quantitatively measure the inclusiveness of spatial relationships. We first define the samples,

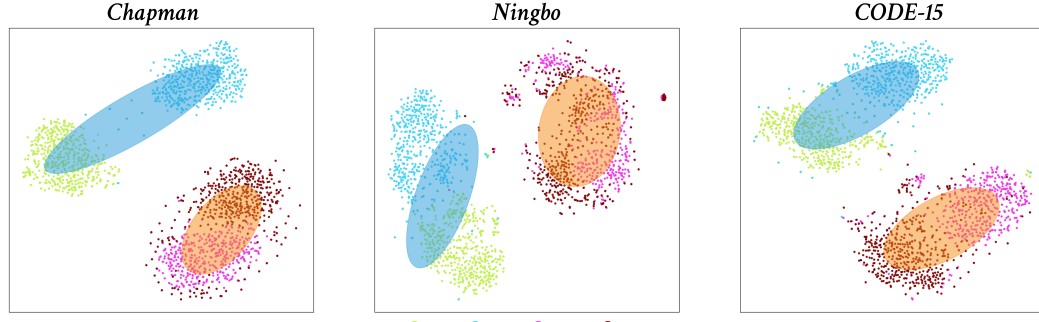

Figure 4: A t-SNE plot of ECG signal representation learned from ST-MEM. Each circle represents the single ECG signal representation with different leads. The ellipse with blue and orange indicates the Gaussian (i.e., a cluster) obtained from the Gaussian mixture model (GMM).

Table 4: Accuracy of representation clustering on two different groups: limb and precordial leads. Each model provides all representations of ECG signals included in *Chapman* through a pre-trained encoder. Representations are clustered using the Gaussian mixture model (GMM). After clustering, each score represents the accuracy of whether the clusters are accurately grouped into limb and precordial leads. 12-lead ECG signals are used in this experiment.

|  | *Chapman* | | |
| --- | --- | --- | --- |
|  | ST-MEM (Ours) | MTAE | MoCo v3 |
| Accuracy | **0.895** | 0.607 | 0.506 |

assigned to a Gaussian, as predicted limb or precordial lead labels. Then, we compute the accuracy of whether the lead sample is correctly clustered into a frontal (i.e., limb leads) or horizontal (i.e., precordial leads) plane. Since the baseline methods, MTAE and MoCo v3, could not consider spatial information, each representation of the ECG signal is randomly assigned to a cluster that causes low accuracy. However, ST-MEM shows high accuracy since samples are grouped in the frontal or horizontal plane.

## 5.5 ABLATION STUDY

Table 5 shows the importance of the lead indication module for ST-MEM. We conduct the linear evaluation in both *PTB-XL* and *CPSC2018* datasets and compute the AU-ROC score. Applying only spatio-temporal patchifying without considering the lead type yields the lowest performance. However, when we include lead-wise shared encoder and each lead indication module (e.g., SEP embedding, and lead embedding), the results of linear evaluation monotonically increase in a *CPSC2018* dataset. This

Table 5: Effectiveness of lead-wise indication module. Each score indicates the AUROC score.

| Method | *PTB-XL* | *CPSC2018* |
| --- | --- | --- |
| Spatiotemporal patchifying | 0.805 | 0.861 |
| & Lead-wise shared decoder | 0.822 | 0.911 |
| & SEP embedding | 0.820 | 0.929 |
| & Lead embedding (Ours) | **0.838** | **0.938** |

demonstrates the effectiveness of spatial information while learning the general ECG representation.

## 5.6 INTERPRETATION OF ST-MEM BY ANALYZING SELF-ATTENTIONS

By analyzing self-attention, we could observe that ST-MEM considers the ECG signal spatially and temporally. Figure 5 is an attention map for a single lead III query patch (i.e., red dashed box). Since lead III is spatially located in a frontal plane, the attention scores are generally high for limb leads. This phenomenon explains why the ECG representation is clustered in the frontal or horizontal plane. Moreover, from a temporal perspective, the patches show high attention scores if the signal shape is similar to the query patch in that the ECG signal contains a periodic rhythm. Overall, we could observe that ST-MEM incorporates both a spatial and temporal relationship of ECG signals. In addition, instead of patchifying temporally or spatially, the spatio-temporal patchifying can also benefit the cardiologist in diagnosing heart disease by analyzing the spatio-temporal self-attentions.

## 6 RELATED WORKS

Our work is closely related to the recent work on self-supervised learning, such as contrastive and generative learning, used widely in computer vision and natural language processing.

**ECG on self-supervised learning.** Recently, research employing deep learning for ECG analysis has extended to a wide range of tasks, including the diagnosis of cardiac diseases such as arrhythmia classification (Ribeiro et al., 2020; Hu et al., 2022; Jun et al., 2018; Strodthoff et al., 2020), emotion recognition (Sarkar & Etemad, 2020), and patient identification (Oh et al., 2022; Li et al., 2020). Furthermore, self-supervised learning is utilized to learn general representations, with some studies focusing on contrastive learning methods. In contrastive learning, some papers apply concepts such as semantic preservation after data augmentation (Chen et al., 2020; He et al., 2020), to ECG data (Lai et al., 2023). Additionally, there is a study on the utilization of temporal invariance and spatial invariance within a single ECG data to preserve semantics (Kiyasseh et al., 2021). Some research combines CNN and transformer architectures to learn local and global features from ECG data (Oh et al., 2022). In terms of generative learning, MaeFE (Zhang et al., 2022) applies MAE (He et al., 2022), which patchifies temporally or spatially for analysis of ECG. Moreover, some studies (Sawano et al., 2022) were conducted by applying spatio-temporal patchifying, shown in Figure 2 (c), to acquire general ECG representation, but without lead-indicating modules, it is challenging to learn the spatial relationship between leads.

**Challenging on reduced lead ECG.** Obtaining a standard 12-lead ECG, however, can be excessive and often requires high-level clinical knowledge (Giannetta et al., 2020) that may not be readily available. Therefore, recent advancements in ECG technologies have resulted in the creation of smaller and portable devices that can obtain the reduced lead set of 12-lead ECG. This advancement promotes the research to apply a machine learning technique to a reduced lead ECG setting (Kiyasseh et al., 2021; Hannun et al., 2019; Urtnasan et al., 2022; Mathews et al., 2018). It is worth noting that the reduced-lead ECG dataset is less abundant compared to 12-lead ECG, and there is research dedicated to addressing this limitation. In the context of self-supervised learning, RLM (Oh et al., 2022) was introduced to acquire robust representation to input ECG lead combinations. On the other hand, in Knowledge Distillation approaches, there is a work that utilizes a model pre-trained on 12-lead ECG as a teacher model and introduces a student model that takes a single lead as input (Qin et al., 2023). This student model leverages the representation from the 12-lead ECG teacher model.

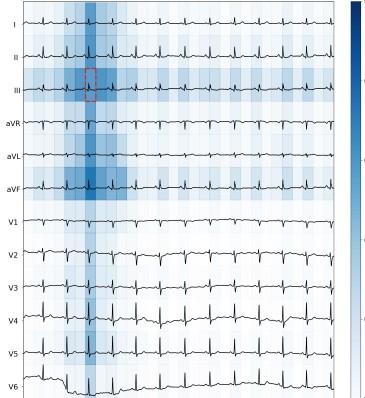

Figure 5: An illustration of an attention map. Attention scores from each encoder layer and head are averaged.

## 7 CONCLUSION

In this work, we propose ST-MEM, Spatio-Temporal Masked Electrocardiogram Modeling, to learn the general ECG representation, generally applicable to diverse ECG problems by incorporating the spatial and temporal relationship of ECG signal. Through extensive experiments, we first demonstrate that ECG representation learned from ST-MEM can exhibit spatial relationships. Since limb leads are measured in the frontal plane, while precordial leads are in the horizontal plane, each representation from ST-MEM reflects that spatial information. Second, we observe the temporal relationship through an attention map. Third, we evaluate the general ECG representation in various experiments, such as reduced lead and low-resource settings. We believe that our work, providing general ECG representation encapsulating the spatio-temporal relationship, can benefit the recent healthcare industry. As part of our future work, we aim to explore the application of ST-MEM to general multivariate time-series datasets. Our experimental results on the human activity recognition (Anguita et al., 2013) task can be found in Appendix C. Moreover, we will explore the various transfer learning approaches (e.g., partially fine-tuning the pre-trained model (He et al., 2022)) to adapt to downstream tasks using general ECG representation instead of naively fine-tuning the model.

## 8 Reproducibility Statement

Our code encompasses the implementation of **ST-MEM** and several other baselines written in Python. Furthermore, we furnish the pre-trained model parameters to facilitate others in fine-tuning the model and achieving reproducible results. Moreover, comprehensive details on training hyperparameters, schemes, and hardware specifications are provided. While we provide just one downstream dataset, *PTB-XL*, already preprocessed and partitioned into train, validation, and test sets, we ensure access to all additional datasets via provided links.

### 8.1 Codes

The codes will be available at `https://github.com/bakqui/ST-MEM`. Please read the 'README.md' file and follow the instructions.

### 8.2 Hyperparameters

All details in pre-training, fine-tuning, and linear evaluation can be found in Appendix A.1. It includes basic hyperparameters such as epochs, batch size, and optimizer.

### 8.3 Datasets

In Secion 4.1, there is all the information of datasets including the number of ECGs, the number of patients, sample rate, and its duration.

Moreover, all ECG data are resampled to 250 Hz, and additional information can be found in Appendix A.2. Moreover, one of the downstream datasets, *PTB-XL*, separated into the train, valid, and test datasets, can be downloaded from the link in our codes (README.md).

Although only preprocessed *PTB-XL* is given, all raw data can be downloaded from the below links:

- *PTB-XL*, *Chapman*, *Ningbo* and *CPSC2018* : `https://physionet.org/content/challenge-2021/1.0.3/`
- *CODE-15* : `https://zenodo.org/record/4916206#.YUG9MStxeUl`
- *PhysioNet2017* : `https://physionet.org/content/challenge-2017/1.0.0/`

*Ningbo* and *Chapman* can be downloaded together in PhysioNet Challenge 2021 (45,152 ECG recordings). For *CODE-15*, as we mentioned in Secion 4.1, ECGs with a duration of less than 10 seconds should be dropped out, and there are some zero padding at the start and end of the recordings, these zero-paddings should be stripped out.

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

# A  DETAILS IN EXPERIMENTAL SETTINGS

In this section, we provide in-depth explanations of our experimental setup, including our approach to dataset processing, model evaluation on downstream tasks, and implementation of baseline methodologies.

## A.1  IMPLEMENTATION DETAILS

All backbones are fixed as ViT-B with patch size 75 and different patch projection layers corresponding to the patchifying way as shown in Figure 2; temporal patchifying is adopted in MoCo v3, CMSC, and MTAE pre-training; spatial patchifying is used in MLAE pre-training; ST-MEM and its variants for ablation study use spatio-temporal patchifying. We use fixed sinusoid temporal positional embedding, while other special embeddings (e.g. mask, lead, and SEP embedding) are learned during training.

For contrastive pre-training, InfoNCE (Oord et al., 2018) loss is used for MoCo v3 and CMSC pre-training. In CMSC pre-training, we divide ECG signals with length $T$ into two temporal non-overlapped segments with length $T/2$ to make positive pairs. In MTAE, MLAE, and ST-MEM pre-training, we mask 75% randomly selected patches and reconstruct them. Mean squared error loss is used for these generative pre-training. We fix the decoder design for all masked auto-encoder variants as four transformer blocks, four heads, and 256 width. For RLM, we randomly mask each lead with probability 0.5. Further details of hyperparameters used in each pre-training is shown in Table 6.

Table 6: Hyperparameter settings.

|  | Pre-training | Fine-tuning | Linear evaluation |
|---|---|---|---|
| Backbone | ViT-B | ViT-B | ViT-B |
| Learning rate | 0.0012 | 0.001 | 0.001 |
| Batch size | 2048 | 1024 | 32 |
| Epochs | 800 | 100 | 100 |
| Optimizer | AdamW | AdamW | AdamW |
| Learing rate scheduler | Cosine anealing | Cosine anealing | Cosine anealing |
| Warump steps | 40 | 5 | 5 |

For environment details, all experiments examined with Ubuntu 20.04.6, AMD EPYC 7502 32-Core Processor, and NVIDIA GeForce RTX 3080 Ti. The version of the libraries we used in all experiments are 3.9.13 for Python and 1.11.0 for PyTorch.

## A.2  DATA PREPROCESSING

**Describing target labels of each downstream dataset.** As mentioned earlier, we have three downstream datasets: *PTB-XL*, *CPSC2018*, and *PhysioNet2017*. In Table 7, there are 44 diagnostic labels for *PTB-XL*, each with its corresponding description. These 44 labels are merged into 23 subclass labels, and these 23 subclass labels are further merged into 5 superclass labels (Wagner et al., 2020). The 5 superclass labels are Myocardial Infarction (MI), Conduction Disturbance (CD), ST/T-Change (STTC), Hypertrophy (HYP), and Normal ECG (NORM). These 5 labels are our target labels in *PTB-XL*. Table 8 provides explanations for the 9 arrhythmia target labels in *CPSC2018* (Liu et al., 2018), while Table 9 contains descriptions for the 4 target labels in the *PhysioNet2017* dataset (Clifford et al., 2017).

**ECG signal processing** We have 6 ECG datasets, each with different characteristics. For instance, *Chapman* and *CODE-15* have varying sampling frequencies (500Hz and 400Hz) and units (microvolts and millivolts). These differences in data resolution and scale could potentially confuse the model and diminish prediction performance. To ensure consistency among datasets and improve ECG signal quality, we implemented several signal processing procedures. Initially, we resampled all ECGs to a uniform sampling frequency of 250Hz. Next, we applied a bandpass digital filter

with a range of 0.67-40Hz to eliminate baseline wandering and high-frequency noise. Finally, we normalized the ECG signals using Z-normalization to ensure uniform scales. The signal processing was conducted for all baselines regardless of whether it was during the pre-training or fine-tuning stage.

**Cropping ECGs into disjoint 10-second segments.** *Chapman*, *Ningbo*, *CODE-15* and *PTB-XL* consist of 10-second ECG data, except for *CPSC2018* and *PhysioNet2017*. To standardize all ECG data to 10 seconds, data with a duration of less than 10 seconds are discarded, and only data exceeding 10 seconds are utilized. For those ECGs more than 10 seconds, they are cropped into non-overlapping 10-second segments. Each crop is evaluated individually and tagged with an origin label for later use in loss computation (Zhang et al., 2022; Gopal et al., 2021; Lan et al., 2022).

**Removing ECGs which have more than one label for multi-class classification setting.** A single ECG may exhibit multiple heart diseases simultaneously, i.e., one ECG could show both MI and CD concurrently. In our ECG datasets, *PTB-XL* and *CPSC2018*, some ECGs have more than one label. In a multi-class classification setting, each ECG should have only one label. Consequently, we excluded ECGs with more than one concurrent label (Zhang et al., 2022).

**Dividing downstream datasets into train, validation and test set.** Finally, regarding the downstream datasets, they are divided into training, validation, and test sets, following a 70-10-20 configuration. Table 10 provides the preprocessing steps for *PTB-XL*, along with information about the utilized train, validation, and test sets. Likewise, Table 11 presents information regarding *CPSC2018*, while Table 12 outlines details concerning *PhysioNet2017*.

## A.3 EVALUATION ON DOWNSTREAM TASKS

The downstream task involves classifying arrhythmias and myocardial infarction. Due to imbalances in the labels within these downstream datasets, we assess model performance using metrics beyond accuracy, including F1 score and AUROC (Area Under the Receiver Operating Characteristic curve) for each experiment. Furthermore, since there are more than two classes involved, we compute the average of one-vs-rest AUROC and the macro F1-score to provide a comprehensive evaluation of the model's performance across multiple classes. As we mentioned before, all cropped ECG segments are evaluated individually, neither averaging nor voting.

Table 7: Description of each diagnostic label of *PTB-XL*.

| Label | Description | Subclass | Superclass |
|---|---|---|---|
| LAFB | left anterior fascicular block | LAFB/LPFB | CD |
| IRBBB | incomplete right bundle branch block | IRBBB | CD |
| AVB | first degree AV block | _AVB | CD |
| IVCD | non-specific intraventricular conduction disturbance (block) | IVCD | CD |
| CRBBB | complete right bundle branch block | CRBBB | CD |
| CLBBB | complete left bundle branch block | CLBBB | CD |
| LPFB | left posterior fascicular block | LAFB/LPFB | CD |
| WPW | Wolff-Parkinson-White syndrome | WPW | CD |
| ILBBB | incomplete left bundle branch block | ILBBB | CD |
| 3AVB | third degree AV block | _AVB | CD |
| 2AVB | second degree AV block | _AVB | CD |
| LVH | left ventricular hypertrophy | LVH | HYP |
| LAO/LAE | left atrial overload/enlargement | LAO/LAE | HYP |
| RVH | right ventricular hypertrophy | RVH | HYP |
| RAO/RAE | right atrial overload/enlargement | RAO/RAE | HYP |
| SEHYP | septal hypertrophy | SEHYP | HYP |
| IMI | inferior myocardial infarction | IMI | Ml |
| ASMI | anteroseptal myocardial infarction | AMI | Ml |
| ILMI | inferolateral myocardial infarction | IMI | Ml |
| AMI | anterior myocardial infarction | AMI | Ml |
| ALMI | anterolateral myocardial infarction | AMI | Ml |
| INJAS | subendocardial injury in anteroseptal leads | AMI | Ml |
| LMI | lateral myocardial infarction | LMI | Ml |
| INJAL | subendocardial injury in anterolateral leads | AMI | Ml |
| IPLMI | inferoposterolateral myocardial infarction | IMI | Ml |
| IPMI | inferoposterior myocardial infarction | IMI | Ml |
| INJIN | subendocardial injury in inferior leads | IMI | Ml |
| PMI | posterior myocardial infarction | PMI | Ml |
| INJLA | subendocardial injury in lateral leads | AMI | Ml |
| INJIL | subendocardial injury in inferolateral leads | IMI | Ml |
| NORM | normal ECG | NORM | NORM |
| NDT | non-diagnostic T abnormalities | STTC | STTC |
| NST_ | non-specific ST changes | NST_ | STTC |
| DIG | digitalis-effect | STTC | STTC |
| LNGQT | long QT-interval | STTC | STTC |
| ISC_ | non- specific ischemic | ISC_ | STTC |
| ISCAL | ischemic in anterolateral leads | ISCA | STTC |
| ISCIN | ischemic in inferior leads | ISCI | STTC |
| ISCIL | ischemic in inferolateral leads | ISCI | STTC |
| ISCAS | ischemic in anteroseptal leads | ISCA | STTC |
| ISCLA | ischemic in lateral leads | ISCA | STTC |
| ANEUR | ST-T changes compatible with ventricular aneurysm | STTC | STTC |
| EL | electrolytic disturbance or drug (former EDIS) | STTC | STTC |
| ISCAN | ischemic in anterior leads | ISCA | STTC |

Table 8: Description of each label of *CPSC2018*.

| Label | Description |
|---|---|
| NORMAL | normal ECG |
| AF | atrial fibrillation |
| 1AVB | first-degree atrioventricular block |
| LBBB | left bundle branch block |
| RBBB | right bundle branch block |
| PAC | premature atrial contraction |
| PVC | premature ventricular contraction |
| STD | ST-segment depression |
| STE | ST-segment elevated |

Table 9: Description of each label of *PhysioNet2017*.

| Label | Description |
|---|---|
| NORMAL | normal sinus rhythm |
| AF | atrial fibrillation |
| OTHER RHYTHM | alternative rhythm |
| NOISY | too noisy to be classified |

Table 10: The number of instances (ECGs) of downstream dataset *PTB-XL*.

| | # ECGs | NORMAL | MI | STTC | CD | HYP |
|---|---|---|---|---|---|---|
| Original | 21837 | 9528 | 5486 | 5250 | 4907 | 2655 |
| After removing ECGs | 16272 | 9083 | 2538 | 2406 | 1709 | 536 |
| Train | 11390 | 6379 | 1752 | 1706 | 1177 | 376 |
| Valid | 3255 | 1797 | 507 | 460 | 373 | 118 |
| Test | 1627 | 907 | 279 | 240 | 159 | 42 |

Table 11: The number of instances (ECGs) of downstream dataset *CPSC2018*.

| | # ECGs | NORMAL | AF | 1AVB | LBBB | RBBB | PAC | PVC | STD | STE |
|---|---|---|---|---|---|---|---|---|---|---|
| Original | 6877 | 918 | 1098 | 704 | 207 | 1695 | 574 | 653 | 826 | 202 |
| After cropping ECGs | 9364 | 1201 | 1593 | 900 | 300 | 2357 | 1014 | 1256 | 1102 | 332 |
| After removing ECGs | 8682 | 1201 | 1266 | 840 | 225 | 1911 | 892 | 1104 | 971 | 272 |
| Train | 6077 | 849 | 865 | 585 | 161 | 1336 | 636 | 779 | 673 | 193 |
| Valid | 868 | 113 | 131 | 84 | 18 | 212 | 85 | 113 | 86 | 26 |
| Test | 1737 | 239 | 270 | 171 | 46 | 363 | 171 | 212 | 212 | 53 |

Table 12: The number of instances (ECGs) of downstream dataset *PhysioNet2017*.

| | # ECGs | Normal | AF | OTHER RHYTHM | NOISY |
|---|---|---|---|---|---|
| Original | 8528 | 5154 | 771 | 2557 | 46 |
| After cropping ECGs | 26940 | 15765 | 2317 | 8237 | 621 |
| Train | 18771 | 10946 | 1608 | 5778 | 439 |
| Valid | 2710 | 1537 | 217 | 875 | 81 |
| Test | 5459 | 3282 | 492 | 1584 | 101 |

## B ADDITIONAL EXPERIMENTS AND RESULTS

### B.1 COMPARISON OF EXISTING ECG PRE-TRAINED MODELS WITH ST-MEM

While we reproduced pre-trained baselines for comparison with ST-MEM, it is important to note that the pre-trained models may exhibit slight variations due to heterogeneous experimental settings, including diverse datasets, backbone encoder models, and hyperparameters. To further explore this, we conduct additional experiments utilizing another pre-trained model, Contrastive Predictive Coding (CPC) (Mehari & Strodthoff, 2022), which provides original pre-trained weights. As shown in Table 13, our proposed approach, ST-MEM, demonstrates comparable performance to CPC. Moreover, in low-resource settings, certain baseline models like MLAE and MoCo v3 exhibit similar performance, emphasizing the comparability of our reproduced pre-trained models as baselines. Additionally, it is worth mentioning that the CPC model employed *PTB-XL* and *CPSC2018* as pre-trained datasets, whereas ST-MEM and other baselines treated these datasets as unseen datasets to validate the genuine performance of general ECG representation learning. Overall, we believe ST-MEM can provide general ECG representation to effectively solve diverse challenging tasks, e.g., diverse heart disease classification tasks, reduced lead settings, and low-resource settings.

Table 13: Experiments in low-resource settings. CPC[†] (Mehari & Strodthoff, 2022) indicates the original implementation of pre-trained model that utilizes pre-training datasets including *PTB-XL* and *CPSC2018*. 250 Hz and 100 Hz represent the sampling rate for ECG preprocessing during the fine-tuning stage. Note that CPC was pre-trained in a sample rate of 100 Hz, yet our default sample rate experimental setting is 250 Hz. Furthermore, 1% and 5% indicate random sampling for training and validation data, while the test data remain constant for all results. Three different samplings were conducted, and the results were averaged. 12-lead ECG signals are used for each result, and the scores represent AUROC scores.

| Methods | PTB-XL | | | CPSC2018 | | |
|---|---|---|---|---|---|---|
| | 1% | 5% | 100% | 1% | 5% | 100% |
| Supervised | $0.676 \pm 0.011$ | $0.736 \pm 0.020$ | $0.905 \pm 0.004$ | $0.600 \pm 0.095$ | $0.609 \pm 0.111$ | $0.958 \pm 0.002$ |
| MoCo v3 | $0.797 \pm 0.006$ | $0.826 \pm 0.015$ | $0.913 \pm 0.002$ | $0.791 \pm 0.045$ | $0.903 \pm 0.019$ | $0.967 \pm 0.003$ |
| CMSC | $0.648 \pm 0.064$ | $0.773 \pm 0.023$ | $0.877 \pm 0.003$ | $0.625 \pm 0.013$ | $0.732 \pm 0.038$ | $0.938 \pm 0.006$ |
| MTAE | $0.707 \pm 0.024$ | $0.713 \pm 0.001$ | $0.910 \pm 0.001$ | $0.670 \pm 0.032$ | $0.756 \pm 0.013$ | $0.961 \pm 0.001$ |
| MTAE+RLM | $0.730 \pm 0.030$ | $0.730 \pm 0.003$ | $0.911 \pm 0.004$ | $0.708 \pm 0.020$ | $0.726 \pm 0.011$ | $0.960 \pm 0.002$ |
| MLAE | $0.793 \pm 0.007$ | $0.838 \pm 0.018$ | $0.915 \pm 0.001$ | $0.860 \pm 0.013$ | $0.922 \pm 0.007$ | $0.973 \pm 0.002$ |
| (250 Hz) CPC[†] | $0.740 \pm 0.057$ | $0.838 \pm 0.024$ | $0.933 \pm 0.001$ | $0.754 \pm 0.015$ | $0.898 \pm 0.026$ | $0.974 \pm 0.002$ |
| (100 Hz) CPC[†] | $0.773 \pm 0.014$ | $0.842 \pm 0.043$ | $\mathbf{0.934 \pm 0.002}$ | $0.762 \pm 0.058$ | $0.917 \pm 0.016$ | $0.973 \pm 0.003$ |
| ST-MEM (Ours) | $\mathbf{0.815 \pm 0.012}$ | $\mathbf{0.878 \pm 0.011}$ | $0.933 \pm 0.003$ | $\mathbf{0.897 \pm 0.025}$ | $\mathbf{0.952 \pm 0.004}$ | $\mathbf{0.980 \pm 0.001}$ |

### B.2 SELECTING AUGMENTATION FOR PRE-TRAINING MOCO V3

In MoCo v3 pre-training, positive pairs are made by employing eight straightforward time-series augmentations (Nonaka & Seita, 2020):

1. **Erase**: Randomly setting the values of a chosen lead to 0.

2. **Flip**: Randomly inverting the signal vertically.

3. **Drop**: Randomly zeroing out signal values.

4. **Cutout**: Selecting a random interval and setting its values to 0.

5. **Shift**: Randomly shifting the signal temporally.

6. **Sine**: Adding a sine wave to the entire signal.

7. **Partial sine**: Adding a sine wave into a randomly selected interval.

8. **Partial white noise**: Introducing white noise into a randomly chosen interval.

However, certain augmentations, such as flip, shift, sine, and partial sine, may have a negative impact on the semantics of the ECG. Flipping an ECG can result in misalignment with standard lead systems; shifting can lead to a loss of temporal continuity; adding a sine wave or partial sine wave

Table 14: Linear evaluation and fine-tuning performance of MoCo v3 on *PTB-XL* and *CPSC2018*.

| Methods | PTB-XL | | | CPSC2018 | | |
|---|---|---|---|---|---|---|
| | Accuracy | F1 | AUROC | Accuracy | F1 | AUROC |
| *Linear Evaluation* | | | | | | |
| MoCo v3, 4 augmentations | **0.552** | **0.142** | 0.709 | 0.209 | 0.038 | 0.644 |
| MoCo v3, 8 augmentations | **0.552** | **0.142** | **0.739** | **0.268** | **0.08** | **0.712** |
| *Fine-tuning* | | | | | | |
| MoCo v3, 4 augmentations | 0.798 | 0.636 | **0.915** | 0.833 | 0.816 | **0.967** |
| MoCo v3, 8 augmentations | **0.799** | **0.644** | 0.910 | **0.852** | **0.838** | **0.967** |

to an ECG may introduce distortion, altering the original morphology of the signal. Consequently, these changes can lead to the misinterpretation of the ECG, potentially affecting the accuracy of diagnoses and analyses. In terms of contrastive learning, the distortion of semantic information makes it inappropriate to define positive or negative pairs.

Therefore, we pre-train MoCo v3 using the remaining four augmentations (erase, drop, cutout, and partial noise) and compared the results with experiments using the original eight augmentations. After pre-training on the same 12-lead dataset as outlined in the paper, we fine-tune the model on 12-lead ECGs from *PTB-XL* and *CPSC2018*, with the results presented in Table 14. Using all eight augmentations performed better than using only four augmentations for all settings except for fine-tuning the model on PTB-XL. A critical point emphasized here is the challenge of selecting augmentations when working with ECG signals. This complexity highlights our preference for generative learning (masking and reconstructing) over contrastive learning which needs to select and use proper augmentations.

### B.3   EXPERIMENTS OF DIFFERENT PROBLEM SETTINGS FOR DOWNSTREAM DATASET *PTB-XL*

We expand our problem to detect a broader range of heart diseases through multi-label classification for other labels in *PTB-XL*. *PTB-XL* consists of a total of 71 labels, categorized into diagnostic, form, and rhythm. As mentioned earlier, the diagnostic labels consist of 44 labels, which are merged into 23 subclasses, with their explanations provided in Table 7. Additionally, there are 19 form labels and 12 rhythm labels, described in Table 15 and Table 16, respectively.

We conducted multi-label classification for these three different settings, following a similar process as in the preprocessing steps described in Appendix A, excluding the step of dropping ECGs that have more than one label. We compared our approach, ST-MEM, with baselines such as supervised learning and MTAE. The superiority of our ST-MEM is highlighted in Table 17.

Table 15: Description of each form label of *PTB-XL*.

| Label | Description |
|---|---|
| NDT | non-diagnostic T abnormalities |
| NST₋ | non-specific ST changes |
| DIG | digitalis-effect |
| LNGQT | long QT-interval |
| ABQRS | abnormal QRS |
| PVC | ventricular premature complex |
| STD₋ | non-specific ST depression |
| VCLVH | voltage criteria (QRS) for left ventricular hypertrophy |
| QWAVE | Q waves present |
| LOWT | low amplitude T-waves |
| NT | non-specific T-wave changes |
| PAC | atrial premature complex |
| LPR | prolonged PR interval |
| INVT | inverted T-waves |
| LVOLT | low QRS voltages in the frontal and horizontal leads |
| HVOLT | high QRS voltage |
| TAB | T-wave abnormality |
| STE₋ | non-specific ST elevation |
| PRC(S) | premature complex(es) |

Table 16: Description of each rhythm label of *PTB-XL*.

| Label | Description |
|---|---|
| SR | sinus rhythm |
| AFIB | atrial fibrillation |
| STACH | sinus tachycardia |
| SARRH | sinus arrhythmia |
| SBRAD | sinus bradycardia |
| PACE | normal functioning artificial pacemaker |
| SVARR | supraventricular arrhythmia |
| BIGU | bigeminal pattern (unknown origin, SV or Ventricular) |
| AFLT | atrial flutter |
| SVTAC | supraventricular tachycardia |
| PSVT | paroxysmal supraventricular tachycardia |
| TRIGU | trigeminal pattern (unknown origin, SV or Ventricular) |

Table 17: Fine-tuning average AUROC results for different settings of *PTB-XL*.

| Methods | Categories | | |
|---|---|---|---|
| | Subclass | Form | Rhythm |
| Supervised | $0.914 \pm 0.002$ | $0.829 \pm 0.021$ | $0.934 \pm 0.003$ |
| MTAE | $0.911 \pm 0.001$ | $0.793 \pm 0.014$ | $0.920 \pm 0.005$ |
| ST-MEM (Ours) | $\mathbf{0.929 \pm 0.001}$ | $\mathbf{0.895 \pm 0.008}$ | $\mathbf{0.966 \pm 0.004}$ |

## B.4 ADDITIONAL SINGLE LEAD EXPERIMENTS ON VARIOUS LEAD TYPES

Mobile devices such as smartwatches usually provide lead I ECGs. However, several studies (Li et al., 2022; Han et al., 2021) show the possibility of obtaining other lead types of single lead ECGs. Therefore, as shown in Table 18, we demonstrate the additional single lead experiments. Interestingly, ST-MEM still outperforms other baselines, MTAE+RLM and MLAE. Moreover, in lead II and aVL, our proposed approach, ST-MEM, surpasses the baselines with a non-trivial margin. This shows that ST-MEM is not only applicable to 12-lead standard ECGs but also to single lead ECGs.

Table 18: Results of single lead on *PTB-XL*. Each value indicates the AUROC score.

| Method | PTB-XL | | | | | |
|---|---|---|---|---|---|---|
| | Lead I | Lead II | Lead III | Lead aVR | Lead aVL | Lead aVF |
| MTAE+RLM | 0.795 ± 0.003 | 0.833 ± 0.002 | 0.742 ± 0.005 | 0.839 ± 0.002 | 0.753 ± 0.006 | 0.801 ± 0.003 |
| MLAE | 0.797 ± 0.001 | 0.826 ± 0.003 | 0.743 ± 0.006 | 0.830 ± 0.003 | 0.753 ± 0.004 | 0.793 ± 0.002 |
| ST-MEM (Ours) | **0.804 ± 0.005** | **0.856 ± 0.002** | **0.788 ± 0.019** | **0.840 ± 0.003** | **0.805 ± 0.003** | **0.819 ± 0.022** |

## B.5 RESULTS OF SINGLE LEAD ECGS IN A LOW-RESOURCE SETTING

While smart devices readily supply unlabeled single lead ECGs, we believe acquiring a small number of labeled ECGs from cardiologists is a feasible endeavor. As demonstrated in Table 19, we show the results of single lead ECGs in low-resource settings. Leveraging the capabilities of general ECG representation learning, ST-MEM consistently surpasses the performance of the supervised baseline model. Notably, ST-MEM achieves an AUROC score of 0.786 using only 5% of lead II ECGs.

Table 19: Experiments of low-resource settings using single lead ECGs. 1% and 5% indicate the random sampling for training and validation data; however, the test data are the same for all results. Three different sampling was done, and the results were averaged. The score represents the AUROC scores.

| Method | PTB-XL | | | | | |
|---|---|---|---|---|---|---|
| | Lead I | Lead II | Lead III | Lead aVR | Lead aVL | Lead aVF |
| | 1% | | | | | |
| Supervised | 0.638 ± 0.035 | 0.657 ± 0.025 | 0.538 ± 0.05 | 0.668 ± 0.021 | 0.562 ± 0.015 | 0.603 ± 0.03 |
| ST-MEM (Ours) | **0.673 ± 0.012** | **0.752 ± 0.03** | **0.640 ± 0.008** | **0.681 ± 0.008** | **0.667 ± 0.006** | **0.674 ± 0.01** |
| | 5% | | | | | |
| Supervised | 0.655 ± 0.018 | 0.682 ± 0.006 | 0.577 ± 0.041 | 0.654 ± 0.053 | 0.593 ± 0.003 | 0.652 ± 0.012 |
| ST-MEM (Ours) | **0.694 ± 0.017** | **0.786 ± 0.024** | **0.700 ± 0.016** | **0.691 ± 0.012** | **0.718 ± 0.026** | **0.723 ± 0.024** |

## B.6 ADDITIONAL ANALYSIS OF ECG REPRESENTATION LEARNED FROM ST-MEM

In order to validate the effectiveness of the lead indication module (e.g., lead-wise shared decoder, SEP embedding, and lead embedding), we plot t-SNE for sampled 12-lead ECG signals across regular and irregular rhythms which are obtained from a *Chapman* dataset. As shown in Figure 6 (a) and (c), the representation of ECG signals provided from ST-MEM clusters the lead embedding by limb leads and precordial leads. However, as depicted in Figure 6 (b) and (d), a model that excludes all lead indication modules struggles to cluster ECG representation.

## B.7 ADDITIONAL ABLATION STUDY

We extend our analysis through an additional ablation study within our framework. In this study, we vary key pre-training stage parameters to understand their impact on the representation. We assess how these changes influence the downstream fine-tuning performance, specifically within the *PTB-XL* dataset.

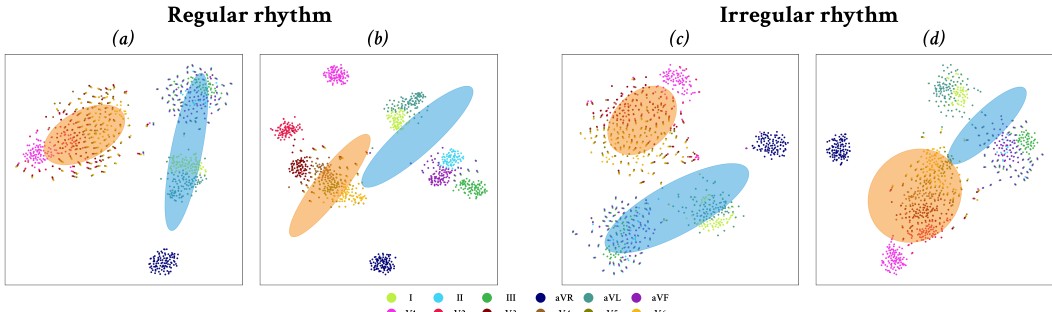

Figure 6: A t-SNE plot of ECG signal representation for the regular and irregular rhythm learned from ST-MEM. Each circle represents the single ECG signal representation with different leads. The ellipse with blue and orange indicates the Gaussian (i.e., a cluster) obtained from the Gaussian mixture model (GMM). (a) and (c) are our proposed method, ST-MEM, where (b) and (d) are the model that removes all lead indication modules (i.e., excluding shared decoder, lead embedding, and SEP embedding).

**Decoder depth.** First, we explore the impact of the depth of the lead-wise shared decoder. Modifying the depth of decoder can influence the characteristics of the representations learned by the encoder. For instance, if the decoder is too shallow, some parts of the encoder might end up taking over the role of decoder (Park et al., 2022). Table 20 provides insights into the impact of varying the decoder depth on *PTB-XL* fine-tuning performance. Notably, when configuring the decoder with just a single transformer block, a significant decrease in performance is observed. Conversely, excessive depth is also found to be undesirable. The highest performance is achieved with a depth of 4 blocks.

Table 20: Ablation on decoder depth.

| Decoder depth | AUROC |
|---|---|
| 1 | 0.925 |
| 4 | **0.933** |
| 8 | 0.931 |
| 11 | 0.931 |

**Masking ratio.** Next, we examine the impact of the random masking ratio during pre-training. The masking ratio directly influences the number of unmasked patches accessible to the decoder, a factor that governs the complexity of the reconstruction task. Figure 7 illustrates how *PTB-XL* fine-tuning performance changes with varying masking ratios. Notably, for ST-MEM, the highest performance is achieved at a moderately high masking ratio of 75%.

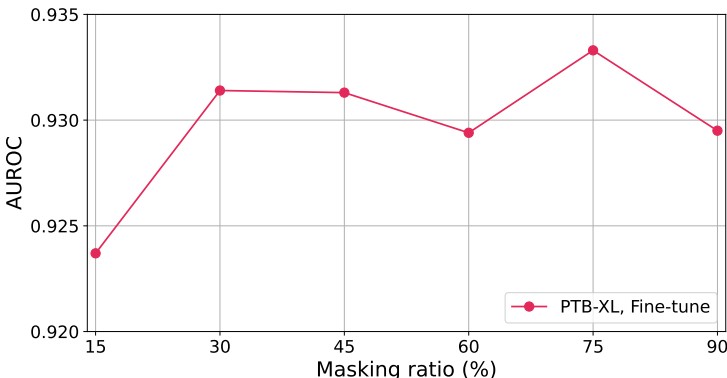

Figure 7: Ablation on masking ratio.

**Lead indicating modules.** We compare the effectiveness of the lead indicating embeddings, SEP and lead embeddings, by removing them one by one. Table 21 is the *PTB-XL* linear evaluation performances of the model trained with or without each lead indicating module. The AUROC score is the highest when both two modules are hired together, meaning that they help the model to learn spatio-temporal relationships.

Table 21: Ablation on lead indicating modules.

| Ablation | | AUROC |
|---|---|---|
| [SEP] embedding | Lead embedding | |
| ✗ | ✗ | 0.822 |
| ✗ | ✓ | 0.830 |
| ✓ | ✗ | 0.820 |
| ✓ | ✓ | **0.838** |

## B.8 ADDITIONAL ANALYSIS OF SELF-ATTENTION MAPS OF ST-MEM

In Figure 8, we demonstrate additional self-attention maps, generated by pre-trained ST-MEM, from ECGs across different rhythm types. It is notable that, regardless of the rhythm types of ECG, ST-MEM consistently assigns high attention weights to temporal patches that have similar shapes to query patch and neighboring spatial patches. From this observation, we conjecture that the ST-MEM is trained to recognize the morphologies of ECGs while distinguishing the unique characteristics of each lead, which is essential for general ECG representation learning.

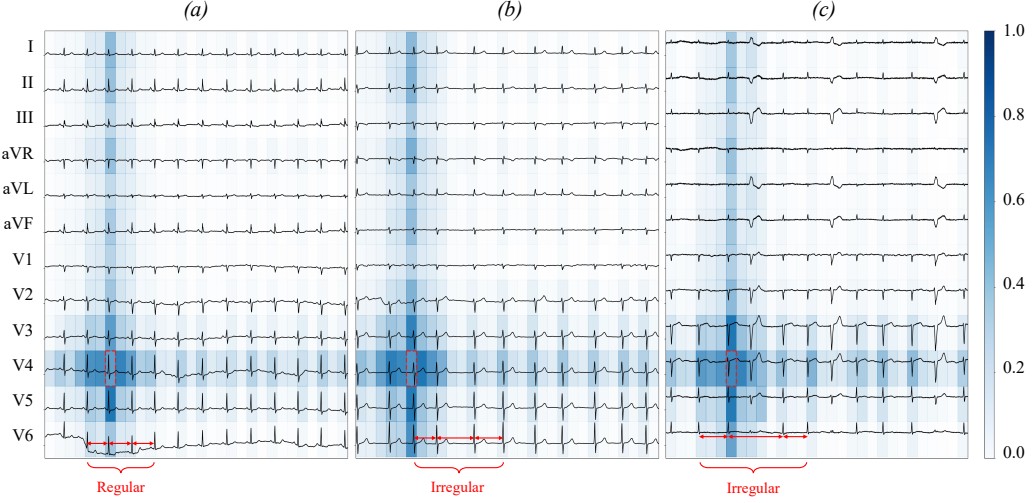

Figure 8: Self-attention maps of ECGs with regular and irregular rhythm. (a) is from a regular rhythm ECG, while (b) and (c) are from irregular rhythm ECGs. Pre-trained ST-MEM consistently assigns high attention weights to temporal patches that share a similar shape to the query patch.

## C   FUTURE WORK: APPLYING ST-MEM TO OTHER MULTIVARIATE TIME-SERIES DATA

We believe that our methodology can be applied to various multivariate time-series domains. An example of this adaptability is the transformation of lead embedding into a general time-series feature embedding. In this section, we demonstrate an additional experiment utilizing the Human Activity Recognition (*HAR*) dataset (Anguita et al., 2013).

*HAR* dataset is a multivariate time series acquired from 30 individuals engaged in daily activities while wearing a waist-mounted smartphone equipped with inertial sensors. Data were collected in 2.56 seconds with a sampling frequency of 50Hz, resulting in 128 readings per sample. The collected data include triaxial acceleration from the accelerometer, estimated triaxial body acceleration, and triaxial angular velocity from the gyroscope, constituting a total of 9 features. The goal is to classify six different activities: walking, walking upstairs, walking downstairs, sitting, standing, and lying down.

The experimental settings are as follows. The model processes input data with dimensions of 9 x 128 with a patch size of 32. The encoder consists of 8 blocks with 64 widths and four heads, while the decoder comprises four blocks with 64 widths and four heads. During pre-training, 60% of patches are randomly masked and reconstructed to minimize the mean squared error between raw signals and reconstruction. The pre-training lasts for 200 epochs with a fixed learning rate of 0.001. Fine-tuning is performed using cross-entropy for 30 epochs with a fixed learning rate of 0.001.

Table 22 is classification results of the fine-tuned model on *HAR*. ST-MEM shows comparable performances, which implies that our proposed model can be extended to the general multi-variate time series domain.

Table 22: Fine-tuning results of human activity classification tasks.

| Methods | *HAR* | |
| --- | --- | --- |
| | Accuracy | F1 |
| Supervised | 0.964 | 0.967 |
| MTAE | 0.965 | 0.968 |
| ST-MEM (Ours) | **0.983** | **0.984** |

