# OpenReview forum: "Guiding Masked Representation Learning to Capture Spatio-Temporal Relationship of Electrocardiogram"
_ICLR.cc/2024/Conference — ICLR 2024 poster_

### Official Review · Reviewer_mpnR · 2023-10-30

**Soundness:** 4 excellent
**Presentation:** 4 excellent
**Contribution:** 3 good
**Rating:** 8
**Confidence:** 5

**Summary:**

The authors propose a simple but effective ECG specific generative self-supervised learning framework utilizing both spatial and temporal characteristics of ECG. The project is straightforward, and well examined.

**Strengths:**

Althogh it's ECG specific, this work provides two important insights. 1) solution for data domain in which cost of labeling is quite high. 2) The electrocardiogram is an indirect observation of the electrical behavior of a single heart from multiple points on the body surface to begin with, and this approach has potential for application to similar data domains that exist not only in medicine.

**Weaknesses:**

The problem setting is specialized to the ECG, and the scope might be too narrow for the general audiencs of ICLR.

**Questions:**

To claim "general ECG representation", it is quite important what actual task was tested. Could you provide more details of classification of myocardial infarction as well as classification of cardiac arrhythmia, e.g., what are the diagnostic criteria of myocardial infarction? What type of arrhythmia are classified?

The authors tested Lead I for the single lead task. In a clinical setting, Lead II is the first choice in many situations. Did the author evaluate for Lead II as well?

It would be quite important for the general audience of ICLR that the authors discuss possible extension of this approach outside of ECG or the medical field.

Minor point.
In Figure 2, the authors assigned the same color code to two different segmentations. Could you change one of them to the other color code such as grayscale?

---

> ### Author Response · Authors · 2023-11-18
>
> ### __[Discussing possible extension of ST-MEM outside of ECGs]__
>
> ㅤ
>
> |   Methods  |     _HAR_    |       |
> |:----------:|:------------:|:-----:|
> |            | __Accuracy__ | __F1__|
> |__Supervised__|        0.964 | 0.967 |
> |  __MTAE__  |        0.965 | 0.968 |
> | __ST-MEM (Ours)__ |      __0.983__|__0.984__|
>
> ㅤ
>
> __Reply__: While our focus is introducing the general ECG representation, we believe that our approach holds the potential for conducting diverse time-series data. Conceptually, the lead can be regarded as a general feature in the time-series domain. To explore this applicability, we demonstrated experiments on human activity recognition [1] (_HAR_) time-series data (as presented in Table 21), comparing the performance of ST-MEM against the supervised model and the masked time autoencoder (MTAE) model. Additionally, we provided an extra section discussing future prospects in Section 8.
>
> ㅤ
>
> __Change__: We included the future work in Section 8. Moreover, we provided the additional experiment on other time-series data, human activity recognition (_HAR_) data, and Appendix J.
>
> ㅤ
>
> ㅤ
>
> ### __[Label information of this paper]__
>
> ㅤ
>
> __Reply__: In Appendix A, we included explanations for all labels in each of the downstream datasets. Table 6, 7, and 8 provide details on the target labels assigned to our three downstream datasets. Multiple arrhythmia such as atrial fibrillation (AFIB), first degree atrioventricular block (1AVB), premature ventricular contraction (PVC) and ST-segment depression (STD) are included as labels. Taking the myocardial infarction (MI) label in _PTB-XL_ as an example, as explained in Table 6, MI is a merged label consisting of 14 individual labels interpreted by cardiologists using the corresponding ECG and the patient's medical record.
>
> ㅤ
>
> __Change__: We updated Appendix A.
>
> ㅤ
>
> ㅤ
>
> ### __[Experiments of different reduced lead settings]__
>
> ㅤ
>
> |  Method  |    _PTB-XL_   |               |               |               |               |               |
> |:--------:|:-------------:|:-------------:|:-------------:|:-------------:|:-------------:|:-------------:|
> |          |   __Lead I__  |  __Lead II__  |  __Lead III__ |  __Lead aVR__ |  __Lead aVL__ |  __Lead aVF__ |
> |__MTAE+RLM__| 0.795 ± 0.003 | 0.833 ± 0.002 | 0.742 ± 0.005 | 0.839 ± 0.002 | 0.753 ± 0.006 | 0.801 ± 0.003 |
> | __MLAE__ | 0.797 ± 0.001 | 0.826 ± 0.003 | 0.743 ± 0.006 | 0.830 ± 0.003 | 0.753 ± 0.004 | 0.793 ± 0.002 |
> | __ST-MEM (Ours)__ |__0.804 ± 0.005__|__0.856 ± 0.002__|__0.788 ± 0.019__|__0.840 ± 0.003__|__0.805 ± 0.003__|__0.819 ± 0.022__|
>
> ㅤ
>
> __Reply__: Thank you for the valuable comment. We only evaluated the lead I in that the lead I is easily accessible through the smartwatch. However, as you suggested, we provided additional single-lead experiments on the other five limb leads in Appendix H, Table 19.
>
> ㅤ
>
> __Change__: We provided Appendix H and Table 19.
>
> ㅤ
>
> ㅤ
>
> ### __[Modifying Figure 2]__
>
> ㅤ
>
> __Reply__: There are some similar comments regarding Figure 2 from the other reviewer as well. Therefore, we decided to modify Figure 2. Hopefully, this illustration can provide you better understanding.
>
> ㅤ
>
> __Change__: Modified Figure 2.
>
> ㅤ
>
> ㅤ
>
>
> ---
> ### __Reference__
>
> ㅤ
>
> [1] Anguita, Davide, et al. "A public domain dataset for human activity recognition using smartphones." Esann. Vol. 3. 2013.

---

> > ### Comment · Reviewer_mpnR · 2023-11-22
> > **Reply to the responses**
> >
> > Thanks for providing additional detals. I thinks these are quite important for the ICLR audiences. I stay "8: accept, good paper".

---

> > > ### Author Response · Authors · 2023-11-22
> > >
> > > We appreciate your effort and time in reviewing our paper. Through your valuable comments, we could have further conducted the experiment on non-ECG time-series data to enhance our paper.

---

### Official Review · Reviewer_QUx7 · 2023-11-01

**Soundness:** 3 good
**Presentation:** 3 good
**Contribution:** 2 fair
**Rating:** 6
**Confidence:** 3

**Summary:**

This paper proposes ST-MEM (Spatio-Temporal Masked Electrocardiogram Modeling) as an approach to learn a feature representation of ECGs which considered both spatial and temporal aspects of the signal. In particular the authors use a lead-specific decoder to guide the network to learn spatial representation. The author demonstrates the effectiveness of the proposed method on multiple datasets includes PTB-XL, CPSC2018 and Physionet 2017 by comparing to supervised learning baseline, as well as other self-supervised learning methods and get superior results on both general setting and low-resource/reduced lead setting. The author also performs quantitative and qualitative analysis of the captured spatial and temporal relationship, with clustering and attention map respectively.

**Strengths:**

1. The paper is well-written with clear methodology and clarity in details. The presentation is readily comprehensible.
2. The proposed method has superior performance on important public datasets compared to several other contrastive and generative self-supervised learning methods. The favorable performance of proposed method also extends to practical scenarios of reduced lead and low-resource setting.

**Weaknesses:**

1. The idea of using MAE to learn representation for ECG signal has been explore previous work(Sawano et al., 2022[1] in the reference). The author should highlight the difference.
2. The experiment session seems missing some previous works. For example, Temesgen et al., 2022[2] also report metrics on PTB-XL and seems to have higher AUROC score.


[1] Shinnosuke Sawano, Satoshi Kodera, Hirotoshi Takeuchi, Issei Sukeda, Susumu Katsushika, and Is- sei Komuro. Masked autoencoder-based self-supervised learning for electrocardiograms to detect left ventricular systolic dysfunction. In NeurIPS 2022 Workshop on Learning from Time Series for Health, 2022.
[2] Mehari, Temesgen, and Nils Strodthoff. "Self-supervised representation learning from 12-lead ECG data." Computers in biology and medicine 141 (2022): 105114.

**Questions:**

1. In the "PERFORMANCE IN REDUCED LEAD SETTINGS" section, 1-lead part was using lead I. How is the choice being made? Would it be valuable to also report 1-lead performance of other 5 leads?
2. In the same section, all methods seem to have larger 6-lead vs 1-lead gap on PTB-XL than CPSC2018. Does the author have insight on why this is the case?

---

> ### Author Response · Authors · 2023-11-18
>
> ### __[Highlighting distinctions from [1] in the methodology]__
> ㅤ
>
> __Reply__: Although we discussed the methodology of this work in Section 6.1, we did not tagged the reference and adequately emphasize the differences from our paper. It solely applies the spatio-temporal patchifying mentioned in Figure 2(c), i.e., excluding the lead indicating modules and lead-wise decoder.
>
> Our comparison with this work is elaborated in Section 5.5. In first row of Table 5, the average AUROC of [1] in _PTB-XL_ and _CPSC2018_ is 0.805 and 0.861, respectively. In contrast, ST-MEM outperforms [1] with average AUROC values of 0.838 and 0.938 in _PTB-XL_ and _CPSC2018_, respectively. We updated Section 6.1 to reflect changes.
>
> Moreover, in Appendix B, by drawing t-SNE plots as Figure 6, we quantitatively analyzed how well the representation was learned. This analysis was conducted on both regular and irregular rhythms. [1], which utilizes only spatio-temporal patchifying, forms clusters effectively within the same lead for (b) and (d), but struggles to form clusters among spatially similar leads. On the contrary, ST-MEM for (a) and (c) not only forms clusters effectively within the same lead but also, limb leads (I, II, III, aVR, aVL, and aVF) cluster together, and precordial leads (V1, V2, V3, V4, V5, and V6) cluster together.
>
> ㅤ
>
> __Change__: We modified Section 6.1.
>
> ㅤ
>
> ㅤ
> ---
> ### __Reference__
>
> [1] Shinnosuke Sawano, Satoshi Kodera, Hirotoshi Takeuchi, Issei Sukeda, Susumu Katsushika, and Is- sei Komuro. Masked autoencoder-based self-supervised learning for electrocardiograms to detect left ventricular systolic dysfunction. In NeurIPS 2022 Workshop on Learning from Time Series for Health, 2022.

---

> ### Author Response · Authors · 2023-11-18
>
> ### __[Comparison with other previous work]__
>
> ㅤ
>
> #### __Table caption__: Experiments in low-resource settings. CPC indicates the original implementation of pre-trained model that utilizes pre-training datasets including _PTB-XL_ and _CPSC2018_. 250 Hz and 100 Hz represent the sampling rate for ECG preprocessing during the fine-tuning stage. Note that CPC was pre-trained in a sample rate of 100 Hz, yet our default sample rate experimental setting is 250 Hz. Furthermore, 1% and 5% indicate random sampling for training and validation data, while the test data remain constant for all results. Three different samplings were conducted, and the results were averaged. 12-lead ECG signals are used for each result, and the scores represent AUROC scores.
> |    Method    |    _PTB-XL_   |               |               |  _CPSC2018_  |               |               |
> |:------------:|:-------------:|:-------------:|:-------------:|:-------------:|:-------------:|:-------------:|
> |              |     __1%__    |     __5%__    |    __100%__   |     __1%__    |     __5%__    |    __100%__   |
> |__Supervised__| 0.676 ± 0.011 |  0.736 ± 0.020 | 0.905 ± 0.004 |  0.600 ± 0.095  | 0.609 ± 0.111 | 0.958 ± 0.002 |
> |  __MoCo v3__ | 0.797 ± 0.006 | 0.826 ± 0.015 | 0.913 ± 0.002 | 0.791 ± 0.045 | 0.903 ± 0.019 | 0.967 ± 0.003 |
> |   __CMSC__   | 0.648 ± 0.064 | 0.773 ± 0.023 | 0.877 ± 0.003 | 0.625 ± 0.013 | 0.732 ± 0.038 | 0.938 ± 0.006 |
> |   __MTAE__   | 0.707 ± 0.024 | 0.713 ± 0.001 |  0.910 ± 0.001 |  0.670 ± 0.032 | 0.756 ± 0.013 | 0.961 ± 0.001 |
> |__MTAE + RLM__|  0.730 ± 0.030  |  0.730 ± 0.003 | 0.911 ± 0.004 |  0.708 ± 0.020 | 0.726 ± 0.011 |  0.960 ± 0.002 |
> |   __MLAE__   | 0.793 ± 0.007 | 0.838 ± 0.018 | 0.915 ± 0.001 |  0.860 ± 0.013 | 0.922 ± 0.007 | 0.973 ± 0.002 |
> |__CPC (250Hz)__| 0.740 ± 0.057 | 0.838 ± 0.024 | 0.933 ± 0.001 | 0.754 ± 0.015 | 0.898 ± 0.026 | 0.974 ± 0.002 |
> |__CPC (100Hz)__| 0.773 ± 0.014 | 0.842 ± 0.043 | __0.934 ± 0.002__ | 0.762 ± 0.058 | 0.917 ± 0.016 | 0.973 ± 0.003 |
> |  __ST-MEM (Ours)__  | __0.815 ± 0.012__ | __0.878 ± 0.011__ | 0.933 ± 0.003 | __0.897 ± 0.025__ | __0.952 ± 0.004__ |  __0.980 ± 0.001__ |
>
> ㅤ
>
> __Reply__: The paper [2] proposes Contrastive Predictive Coding (CPC) to ECG data. However, the experimental setting for [2] is different from ours which causes the discrepancy in the AUROC score in the _PTB-XL_ dataset. For instance, their problem setting is a multi-label classification for all 71 labels; however, we define the problem as multi-class classification using super-class labels provided from the original _PTB-XL_ paper. Our detailed experimental settings can be found in Appendix A.1. Sorry for the unclearness of the experimental setting.
>
> Since the experimental setting is different from [2], we utilized the pre-trained weights provided in the original paper. Then, we fine-tuned the CPC model on our settings. We could validate that ST-MEM still shows comparable performance and surpasses all the low-resource settings even though the CPC model utilized _PTB-XL_ and _CPSC2018_ to pre-train the model. We provided the additional experiments in Appendix K.
>
> ㅤ
>
> __Change__: We added the new section as Appendix K.
>
> ㅤ
>
> ㅤ
>
> ---
> ### __Reference__
>
> [2] Mehari, Temesgen, and Nils Strodthoff. "Self-supervised representation learning from 12-lead ECG data." Computers in biology and medicine 141 (2022): 105114.

---

> ### Author Response · Authors · 2023-11-18
>
> ### __[Experiments of different reduced lead settings]__
> ㅤ
>
> |  Method  |    _PTB-XL_   |               |               |               |               |               |
> |:--------:|:-------------:|:-------------:|:-------------:|:-------------:|:-------------:|:-------------:|
> |          |   __Lead I__  |  __Lead II__  |  __Lead III__ |  __Lead aVR__ |  __Lead aVL__ |  __Lead aVF__ |
> |__MTAE+RLM__| 0.795 ± 0.003 | 0.833 ± 0.002 | 0.742 ± 0.005 | 0.839 ± 0.002 | 0.753 ± 0.006 | 0.801 ± 0.003 |
> | __MLAE__ | 0.797 ± 0.001 | 0.826 ± 0.003 | 0.743 ± 0.006 | 0.830 ± 0.003 | 0.753 ± 0.004 | 0.793 ± 0.002 |
> | __ST-MEM (Ours)__ | __0.804 ± 0.005__ | __0.856 ± 0.002__ | __0.788 ± 0.019__| __0.840 ± 0.003__ | __0.805 ± 0.003__ | __0.819 ± 0.022__ |
> ㅤ
>
> __Reply__: Thank you for suggesting valuable experiments. Mobile devices such as smartwatches usually provide lead I; thus, we consider the practical scenario to choose lead I for the single lead reduced lead settings. However, as you suggested, we also provided additional single lead experiments in Appendix H. As shown in Table 19, we could validate that ST-MEM outperformed other baselines on other lead types.
>
> ㅤ
>
> __Change__: We provided Appendix H and Table 19.
>
> ㅤ
>
> ㅤ
>
> ### __[Different performance gaps between 6-lead vs 1-lead across dataset]__
>  ㅤ
>
> __Reply__: We think the performance gaps depend on the difficulty of the task. As one can see in Table 3, the overall performance metrics on _CPSC2018_ are higher than _PTB-XL_, which may indicate that the _CPSC2018_ tasks are easier than _PTB-XL_. Specifically, single lead alone may be sufficient to classify particular classes of _CPSC2018_, such as atrial fibrillation (AFIB), premature ventricular complex (PVC), but this may not be the case for certain classes of _PTB-XL_, such as myocardial infarction (MI).
>
> ㅤ
>
> __Change__: None.

---

> ### Comment · Reviewer_QUx7 · 2023-11-23
> **Response to the authors's comment**
>
> Thanks the authors for taking the time to draft clear responses and make the change.
>
> Dear area chairs and Reviewers,
>
> Given the change and clarification from the authors, I am pleased to update the rating to:
> 6: marginally above the acceptance threshold

---

> ### Author Response · Authors · 2023-11-23
>
> Thank you for responding to our comments. We are glad that we could have addressed your concern of the comparison with other related works.

---

### Official Review · Reviewer_knNF · 2023-11-01

**Soundness:** 3 good
**Presentation:** 3 good
**Contribution:** 3 good
**Rating:** 6
**Confidence:** 4

**Summary:**

This paper presents a self-supervised pre-training method for multi-lead ECG data that can be generalized to reduced lead sets. The method relies on spatio-temporal patch reconstruction, with led-wise shared decoders. The experiments were performed by pre-training on three ECG datasets and fine-tuning to two different down-stream ECG datasets/tasks. Results demonstrated the improvement in downstream tasks in comparison to supervised baselines and various unsupervised pre-training methods.

**Strengths:**

The paper was overall clearly written and easy to follow.

The experimentation was thorough, especially in the inclusion of relevant baselines, and the empirical results speak favorably to the contribution of the work.

The use in low-resource settings is interesting, and the gain brought by proposed pre-training more significant.

**Weaknesses:**

Despite favorable empirical results, the intuition of the spatiotemporal patching based self-supervised learning is not very clear. It is not clear why such patching will help the learning of spatiotemporal representations. Furthermore, the relation with two baselines — MLAE that reconstructs spatial patches and MTAE that reconstructs temporal patches — need to be better clarified.

The assumption of the method, in particular to what type of ECG signals and tasks this will apply, needs to be better discussed. For instance, the use of temporal patching seems to rely on the assumption that the rhythm  within the input signal length (e.g., 10s) — whether it is normal or abnormal —is regular and periodic.  Is that true? How would this work for rhythms that are irregular (e.g., atrial fibrillation), or rhythms where the abnormal rhythm only shows up in transient beats within a longer segment (which is typical for some PVCs and tachycardias). If the method is not designed for these scenarios, it should be clearly clarified.

Related to the above comments, more details about the data and classification tasks needed to be given. What are the labels of rhythms being classified? Does each 10-s segment only has one label? The paper did a good job in highlighting the it’s important to develop methodology to the application problem at hand — this should also be reflected when describing data and experimental settings, as different ECG tasks can mean that the features one are looking for is very different (e.g. is it a regular rhythm, is it an irregular rhythm, is it a transient rhythm, etc).

Overall, while the performance gain is notable, they are also overall limited — at 2 decimal points compared to supervised baselines wen using 100% data. It is not clear what is the clinical significance of such performance gain. Perhaps it is because the supervised baselines are already quite good in the tasks considered (over 0.9 AUROC in both downstream datasets). It may be more convincing if the authors could find “harder” base tasks in order to see if the proposed method will have clinical significance.

**Questions:**

Please clarify the main questions listed in my above comments.

In addition, it’d be interesting to see Fig 4, the embedding of the ECG signals, across different rhythm types as well, in order to appreciate the clustering across rhythm types versus spatial locations of the lead.

---

> ### Author Response · Authors · 2023-11-18
>
> ### __[Better explanation of the spatiotemporal patching]__
>
> ㅤ
>
> __Reply__: The spatiotemporal patching allows the encoder to consider both temporal and spatial relationships concurrently. On the other hand, temporal patching leads the encoder to consider only the temporal relationship. The spatial patching (i.e., lead patching) discourages the encoder from capturing temporal relationships. For a better understanding of the patching approach, we modified Figure 2.
>
> ㅤ
>
> __Change__: We modified Figure 2.
>
> ㅤ
>
> ㅤ
>
>
> ### __[Clarifying the baselines, Masked Time Auto Encoder (MTAE) and Masked Lead Auto Encoder (MLAE)]__
>
> ㅤ
>
> __Reply__: Since the MTAE and MLAE take different patching strategies (i.e., temporal or spatial patching), the encoder can only consider either temporal or spatial relationships. In other words, the encoder of MTAE utilizes temporal patching to capture the temporal relationship only, while the encoder of MLAE takes spatial patches to consider the spatial relationship only. Furthermore, MTAE and MLAE cannot explicitly distinguish the lead type information, whereas ST-MEM can encapsulate the lead type information through lead embedding and [SEP] patch embedding. We revised the manuscript to clarify the relation of baselines (MTAE and MLAE) with ST-MEM.
>
> ㅤ
>
> __Change__: We clarified in Section 4.2 that MTAE and MLAE do not have lead indicating modules and differ in their approach to patchifying input ECGs with ST-MEM.
>
> ㅤ
>
> ㅤ
>
>
> ### __[Applicability of the method to regular and irregular ECGs]__
>
> ㅤ
>
> __Reply__: We used the ECGs of both regular and irregular rhythms during pre-training. We assume that the masked parts of the input signal can be reconstructed by utilizing the information extracted from the unmasked portions, regardless of the regularity of the rhythms. We did not mention the detailed characteristics of pre-training data, so we clarified in Section 4.1 that we used all ECGs of pre-training data without focusing on specific labels (e.g., normal rhythm).
>
> Additionally, for more clarification, we demonstrated the self-attention maps of pre-trained ST-MEM for irregular rhythm ECGs in Appendix D. Notably, ST-MEM consistently assigned high attention weights to temporal patches that have similar shape to the query patch and neighboring spatial patches, regardless of the rhythm type of ECG. From this observation, we conjecture that pre-trained ST-MEM can effectively recognize the morphologies of ECG signals.
>
> ㅤ
>
> __Change__: We clarified that we used ECGs of all rhythm types in Section 4.1, and demonstrated the self-attention maps of pre-trained ST-MEM for regular and irregular rhythm ECGs in Appendix D.
>
> ㅤ
>
> ㅤ
>
>
> ### __[Explanation of dataset and classification task]__
>
> ㅤ
>
> __Reply__: In appendix A, we added description for all labels of all 3 downstream datasets. In table 6, 7, and 8, we have outlined the target labels for our three downstream datasets. As apparent from the tables, we employed a variety of ECGs in both training and evaluation, encompassing regular rhythms such as Normal ECG and irregular rhythms like atrial fibrillation (AFIB), first degree atrioventricular block (1AVB) and premature ventricular contraction (PVC). Notably, during pretraining, we used all ECGs without focusing on specific labels. Typically, an ECG may have more than one label, but for our multiclass classification setting, we used only ECGs with a single label, discarding the rest. This processing step is also depicted in Section 4.1 and Appendix A.
>
> ㅤ
>
> __Change__: We modified Section 4.1 and Appendix A.

---

> ### Author Response · Authors · 2023-11-18
>
> ### __[Clinical significance regarding the main result]__
>  ㅤ
>
> |           | Sensitivity | Specificity |
> |-----------|-------------|-------------|
> | __Supervised__| 77.3%       | 90.1%       |
> | __ST-MEM (Ours)__     | __88.2%__       | 90.1%       |
> ㅤ
>
> __Reply__: The increase of AUROC by two decimal points may be interpreted as a slight performance gain. However, given that the AUROC measures the area under the curve representing the trade-off between the true positive rate (i.e., sensitivity) and false positive rate (i.e., 1-specificity), the noteworthy difference in sensitivity at the same specificity becomes quite significant. For instance, we computed the sensitivity and specificity of myocardial infarction (MI) classification results on the _PTB-XL_ dataset using the supervised and ST-MEM model. The supervised baseline provides sensitivity=0.773, whereas ST-MEM achieves 0.882 sensitivity. In other words, ST-MEM can classify 88.2% of MI on ECGs, while the supervised model can only classify 77.3%. We believe this difference is clinically significant since MI is known as a high-mortality disease.
> ㅤ
>
> | _PTB-XL_     |               |               |               |               |               |               |
> |------------|---------------|---------------|---------------|---------------|---------------|---------------|
> | __Method__     | __Lead I__             | __Lead II__            | __Lead III__           | __Lead aVR__           | __Lead aVL__           | __Lead aVF__           |
> |            | __1%__            |               |               |               |               |               |
> | __Supervised__ | 0.638 ± 0.035 | 0.657 ± 0.025 | 0.538 ± 0.050  | 0.668 ± 0.021 | 0.562 ± 0.015 | 0.603 ± 0.030  |
> | __ST-MEM (Ours)__       | __0.673 ± 0.012__ | __0.752 ± 0.030__  | __0.640 ± 0.008__  | __0.681 ± 0.008__ | __0.667 ± 0.006__ | __0.674 ± 0.010__  |
> |            | __5%__            |               |               |               |               |               |
> | __Supervised__ | 0.655 ± 0.018 | 0.682 ± 0.006 | 0.577 ± 0.041 | 0.654 ± 0.053 | 0.593 ± 0.003 | 0.652 ± 0.012 |
> | __ST-MEM (Ours)__       | __0.694 ± 0.017__ | __0.786 ± 0.024__ | __0.700 ± 0.016__   | __0.691 ± 0.012__ | __0.718 ± 0.026__ | __0.723 ± 0.024__ |
> ㅤ
>
> Moreover, considering the label scarcity and challenge of obtaining standard 12 lead ECGs, we provide the additional challenging experiment of single lead in the low-resource setting. As shown in the above table, ST-MEM outperforms the supervised baseline for all single lead types. Interestingly, we could achieve a 0.786 AUROC score with only 5% of lead II ECGs. With the emergence of mobile devices like smartwatches enabling access to single-lead ECG data, we believe this experiment demonstrates the importance of ECG representation learning.
> ㅤ
>
> __Change__: We provided Appendix I and Table 20.
>
> ㅤ
>
> ㅤ
>
>
> ### __[t-SNE plot for ECGs with different rhythm types]__
>  ㅤ
>
> __Reply__: We demonstrated the t-SNE plots only for ECGs with a regular rhythm. We added the t-SNE plots for ECGs with irregular rhythm as well on Appendix B. The appearance of t-SNE plots was similar across different rhythm types, showing that the representations of neighboring leads were gathered closely.
>  ㅤ
>
> __Change__: We added t-SNE plots of ECGs with irregular rhythm in Appendix B.

---

### Official Review · Reviewer_pLfJ · 2023-11-09

**Soundness:** 3 good
**Presentation:** 3 good
**Contribution:** 3 good
**Rating:** 8
**Confidence:** 4

**Summary:**

This work proposes a method called ST-MEM (Spatio-Temporal Masked Electrocardiogram Modeling),
to leverage self-supervised learning (SSL) in order to train a model that can be used for diagnosis of conditions detectable through Electrocardiograms (ECG).

ST-MEM is based on a Masked auto-encoder and it uses a vision transformers (ViT) architecture. The 12 leads ECG signal is divided into temporal patches. Some patches are masked and the task is to reconstruct the masked signal. Each encoded patch is then added to a LEAD-specific encoding (a way to identify from which lead the signal is coming from) and to the traditional positional encoding (a way to identify temporary where the patch belongs). Additionally, a special token is appended and postponed to each lead signal. Since some leads share a highly correlated signal the decoder is a Lead-wise shared decoder (i.e. it only attempts to decode the signal from 1 lead at the time, this is to ensure that the task is not trivially solved by copying the masked patched from a highly correlated lead).

After training a labeled dataset can be sued to train a linear layer on top of the encoder, or to fully fine-tuned the model.

The proposed model is pre-trained using a total of 188,480 ECGs 12 leads signals coming from three datasets (Chapman, Ningbo, CODE-15), and if tested using two different datasets PTB-XL and CPSC2018 on the task of detecting cardiac arrhythmia and myocardial infarction. Results (in terms of accuracy, F1 and AUROC) show that with linear fine-tuning, the proposed method performs better than other SSL methods albeit it still underperforms the supervised baseline. When fine-tuning, however, the proposed method surpasses also the supervised baselined in all the metrics employed.

The current method is also resilient to lower amount of data compared to all alternatives tested as shown by achieving the best results in terms of AUROC using only 1% and 5% of the fine-tuning dataset.

Additionally, the authors perform experiments using only a sub set of the leads or using only 1 lead (as in the PhysioNet2017 dataset). The proposed technique remains the best  performer across all baselines.

**Strengths:**

The paper tackles an important problem since high quality ECG labeled data are scarce but ECG data in general is much more available.

The proposed solution is simple yet very effective as shown in the results, also compared to other SSL alternatives.

**Weaknesses:**

Some part of the manuscript could be improved. For example:
	Figure 2 (c) is not clear.
	What is “seasonality”?
	See more clarification to add to the manuscript from my questions below

Some experiments could be stronger. For example PTB XL dataset have 4 different conditions but it seems that only a couple were used in the tests.

**Questions:**

Having both the SEP token and the leads embedding seems redundant. Have the authors considered an ablation where only the leads embedding are used but not the SEP? If SEP is needed why is it needed twice and not just at the beginning or just at the end?

Have the author considered using only a subset of the 8 augmentations used for the contrastive SSL baselines shown in Appendix D? Some of them could really alter the signal and be counter productive. For example, the Flip, the shift perhaps also the Sine and partial Sine.

The initial statement “detecting various heart diseases” seem to imply that the proposed technique could do so, however, the tests only show a couple of heart condition. For example in the PTB XL dataset there are 4 different conditions, why not show the results on all of them?

“The patches undergo flattening” why is flattening needed here? Isn’t the signal already flat?

The pre-training dataset comes with different sampling rate (two at 500Hz and 1 at 400Hz). How was this taken care of? Was the 500 subsampled? This should be explained.

Similarly the physionet comes with 200Hz, how was it adapted to the pre-trained model?

For signals that are longer than 10 seconds, I assume the signal was split into 10 seconds but how was all the outcome computed? Average? Voting? This should be clarified.

---

> ### Author Response · Authors · 2023-11-18
>
> ### __[Modifying Figure 2]__
>
> ㅤ
>
> __Reply__: Sorry for the unclearness. We modified Figure 2 to illustrate the intuition of different patch strategies.
>
> ㅤ
>
> __Change__: Modified Figure 2.
>
> ㅤ
>
> ㅤ
>
> ### __[Clarifying the word “seasonality” in Section 5.6]__
>
> ㅤ
>
> __Reply__: Sorry for the confusion in Section 5.6. We misused the word "seasonality". We attempted to define the repeatability of QRS segments as the word seasonality since the query patch contains QRS segments. Note that we tried to avoid the use of clinical terms. We modified the sentence and removed the word "seasonality".
>
> ㅤ
>
> __Change__: Revising the sentence in section 5.6.
>
> “Moreover, from a temporal perspective, the ECG signal contains seasonality such that the patch with similar seasonality shows high attention scores.”
>
> ->
>
> “Moreover, from a temporal perspective, the patches show high attention scores if the signal shape is similar to the query patch in that the ECG signal contains a periodic rhythm.”
>
> ㅤ
>
> ㅤ
>
> ### __[Ablation on SEP embedding]__
>
> ㅤ
>
> |    Ablation    |               |_PTB-XL_|
> |:--------------:|:-------------:|:------:|
> | __SEP embedding__ | __Lead embedding__ |        |
> |        X       |       X       |  0.822 |
> |        X       |       O       |  0.830 |
> |        O       |       X       |  0.820 |
> |        O       |       O       |  __0.838__ |
>
> ㅤ
>
> __Reply__: We assessed the effectiveness of the lead indicating modules (e.g., SEP embedding, and lead embedding) by removing them one by one. We added Table 13, the _PTB-XL_ linear evaluation performances of the model trained with or without each lead indicating module in Appendix C. The highest AUROC score is observed when both modules are employed simultaneously, indicating that they mutually contribute to enhancing the model's ability to learn spatio-temporal relationships.
>
> ㅤ
>
> #### __Table caption__: Our proposed case
> |     Patch embedding    |     [SEP]       |     Patch (I)    |     [SEP]       |     [SEP]        |     Patch(II)    |     [SEP]        |
> |------------------------|---------------|------------------|---------------|----------------|------------------|----------------|
> |     __Lead embedding__     |     Lead I    |     Lead I       |     Lead I    |     Lead II    |     Lead II      |     Lead II    |
>
> ㅤ
>
> #### __Table caption__: alternative case
> |     Patch embedding    |     [SEP]       |     Patch (I)    |     [SEP]        |     Patch(II)    |     [SEP]        |
> |------------------------|---------------|------------------|----------------|------------------|----------------|
> |     __Lead embedding__     |     Lead I    |     Lead I       |     Lead II    |     Lead II      |     Lead II    |
>
> ㅤ
>
> Moreover, there is the reason that we appended the SEP embedding twice, i.e., at the beginning and at the end of each lead signal. The above table depicts two cases. Imagine that we do the single lead experiments for lead I ECGs and lead II ECGs. The discrepancy of input shape for lead I and II would discourage the model from distinguishing the lead type. On the other hand, the proposed method can provide a consistent input shape on each single lead experiment. Therefore, we appended SEP embeddings twice.
>
> ㅤ
>
> __Change__: We added Table 13 in Appendix C.

---

> ### Author Response · Authors · 2023-11-18
>
> ### __[Selecting augmentations for pre-training MoCo v3]__
>  ㅤ
> ㅤ
>
> __Reply__: We appreciate your valuable feedback. Some augmentations may not preserve the semantics of the data and even have a negative impact. In that case, in contrastive learning with such augmentations, achieving high-quality representations might be challenging. We excluded four augmentations (flip, shift, sine, and partial sine) that could potentially affect the semantics of ECG data. We then pre-trained MoCo v3 using the remaining four augmentations (erase, drop, cutout, and partial noise) and compared the results with experiments using the original eight augmentations. After pre-training on the same 12-lead dataset, we fine-tuned the model on 12-lead ECGs from _PTB-XL_ and _CPSC2018_. The results are presented in the table below.
>
>
> |     Methods                     |     _PTB-XL_      |              |              |     _CPSC2018_    |              |              |
> |---------------------------------|-----------------|--------------|--------------|-----------------|--------------|--------------|
> |                                 |     __Accuracy__    |     __F1__       |     __AUROC__    |     __Accuracy__    |     __F1__       |     __AUROC__    |
> |     __Linear evaluation__           |                 |              |              |                 |              |              |
> |     __MoCo v3, 4 augmentations__    |   __0.552__     |   __0.142__  |     0.709    |     0.209       |     0.038    |     0.644    |
> |     __MoCo v3, 8 augmentations__    |   __0.552__     |   __0.142__  |   __0.739__  |   __0.268__     |   __0.080__  |   __0.712__  |
> |     __Fine-tuning__                 |                 |              |              |                 |              |              |
> |     __MoCo v3, 4 augmentations__    |     0.798       |     0.636    |     __0.915__    |     0.833       |     0.816    |   __0.967__  |
> |     __MoCo v3, 8 augmentations__    |     __0.799__       |     __0.644__    |     0.910    |     __0.852__       |     __0.838__    |     __0.967__    |
>
>
> As shown in the table, using all eight augmentations performed better than using only four augmentations for all settings except for the fine-tuning the model on _PTB-XL_.
>
>
> Selecting proper augmentations is challenging when working with ECG signals. This complexity emphasizes the need of the generative learning (masking and reconstruction) against the contrastive learning which requires to consider the combination of proper augmentations.
>
> ㅤ
>
> __Change__: We added new section as Appendix G.
>
> ㅤ
>
> ㅤ
>
> ### __[Different problem settings of downstream dataset _PTB-XL_]__
>
> ㅤ
>
> __Reply__: We focused solely on the task of distinguishing the 5 super-classes of diagnostic labels for _PTB-XL_. We conducted additional experiments for the other three settings: 23 diagnostic sub-class labels, 19 form labels, and 12 rhythm labels. In this multi-label classification experiment, our proposed method demonstrated superiority over other baselines (supervised learning and MTAE). Further details have been included in the Appendix F.
>
>
> |     Methods       |     Categories       |                      |                      |
> |-------------------|----------------------|----------------------|----------------------|
> |                   |     __Sub-class__         |     __Form__             |     __Rhythm__           |
> |     __Supervised__    |     0.914 ± 0.002    |     0.829 ± 0.021    |     0.934 ± 0.003    |
> |     __MTAE__          |     0.911 ± 0.001    |     0.793 ± 0.014    |     0.920 ± 0.005    |
> |     __ST-MEM (Ours)__      |     __0.929 ± 0.001__    |     __0.895 ± 0.008__    |     __0.966 ± 0.004__    |
>
> ㅤ
>
> __Change__: We added new section as Appendix F.
>
> ㅤ
>
> ㅤ
>
>
> ### __[Use of word “flattening” in patchfying explanation]__
>
> __Reply__: We are sorry for confusing the audience by misusing the word “flattening.” In spatio-temporal patchifying, in contrast to the temporal patchifying scenario, there is no flatten operation as it is already flattened. We deleted the word “flatten”.
>
> __Change__: Revised the sentence.
>
> “The patches undergo flattening and linear projection to form patch embeddings of dimension D.”
>
> ->
>
> “The patches undergo linear projection to form patch embeddings of dimension D.”

---

> > ### Author Response · Authors · 2023-11-18
> >
> > ### __[Sampling frequency of ECG dataset]__
> >
> > ㅤ
> >
> > __Reply__: For consistency, we resampled all 6 ECG datasets, including pre-training and downstream datasets, to 250Hz. It is explained in Section 4 and Appendix A.1.
> >
> > ㅤ
> >
> > __Change__: We modified Section 4 and Appendix A.1.
> >
> > ㅤ
> >
> > ㅤ
> >
> >
> > ### __[Explanation on evaluation of cropped ECGs]__
> >
> > ㅤ
> >
> > __Reply__: To maintain uniformity, we standardized all ECGs to a duration of 10 seconds. ECGs exceeding 10 seconds were disjointedly cropped into 10-second segments. For instance, if there was a 27-second ECG, we created two individual samples: one from 0 to 10 seconds and another from 10 to 20 seconds, discarding 20 to 27 seconds. These two ECGs were trained and evaluated separately, each retaining the labels of the original 27-second ECG. Therefore, these two ECGs are considered as two individual data points and computed independently, neither averaging nor voting. It is explained in Appendix A.
> >
> > ㅤ
> >
> > __Change__: We modified Section 4.1 and added detail explanation on Appendix A.

---

> > > ### Comment · Reviewer_pLfJ · 2023-11-20
> > >
> > > Thanks to these authors for the additional clarification and results.

---

> > > > ### Author Response · Authors · 2023-11-21
> > > >
> > > > We appreciate your response to our comments. We could have enhanced our paper through your valuable suggestions and comments.

---

### Author Response · Authors · 2023-11-18
**Summary of revisions**

We thank all reviewers for their valuable comments. We appreciate that all reviewers find our work valuable in which we proposed ST-MEM, which is a simple but effective solution for tackling the high labeling cost of ECG by learning general representation, particularly when practical settings such as low-resource condition and reduced lead set settings.

Despite having such strengths, there are commonly asked questions in reviews.

ㅤ

ㅤ

__[Detailed explanation of our dataset and classification task. (Reviewer knNF, Reviewer mpnR)]__

ㅤ

We apologize for the lack of explanation regarding the dataset and our classification task. We clarified this in Appendix A, providing clear details on the datasets we used, their labels, and the processing steps involved.

ㅤ

ㅤ

__[Ambiguity of Figure 2. (Reviewer mpnR, Reviewer pLfJ)]__

ㅤ

We apologize for initially drawing the figure in a confusing manner, so we redrew it. The figure aimed to illustrate three types of patchifying methods. We used different colors for distinct patches. Additionally, there is a query patch with black dashed lines in the three patchifying illustrations, and we used arrows to depict how the self-attention of the Vision Transformer (ViT) operates. As observed in the figure, in the case of (c) spatio-temporal patchifying, it is evident that attention is directed to different time frames of different leads.

ㅤ

ㅤ

__[Utilizing other single leads for experiments of reduced lead setting. (Reviewer mpnR, Reviewer QUx7)]__

ㅤ

In Section 5.3, during the fine-tuning process, we conducted experiments using reduced lead of 12 leads rather than utilizing all 12 leads. Specifically, we chose to use lead I as the single lead in this context.

We acknowledge that cardiologists prioritize lead II when interpreting ECG sheets. However, considering the common usage of mobile devices like smart watches, which typically measure lead I, we chose lead I as the single lead in our experiments.

Additionally, we experimented with a reduced lead setting using single lead from the other limb leads (lead II, lead III, lead aVR, lead aVL, lead aVF). Our proposed method demonstrated superiority compared to other baselines for all the leads. Furthermore, we observed that the lead II model outperformed other single lead models, and this information has been included in Appendix H.

ㅤ

ㅤ

__[Extension to other data domain of ST-MEM. (Reviewer mpnR)]__

ㅤ

While our focus is introducing the general ECG representation, we believe that our approach holds the potential for conducting diverse time-series data. Conceptually, the lead can be regarded as a general feature in the time-series domain. To explore this applicability, we demonstrated experiments on human activity recognition [1] (_HAR_) time-series data in Appendix J. Additionally, we provided an extra section discussing future prospects in Section 8.

ㅤ

ㅤ

---

### __Reference__

ㅤ

[1] Anguita, Davide, et al. "A public domain dataset for human activity recognition using smartphones." Esann. Vol. 3. 2013.

---

> ### Author Response · Authors · 2023-11-21
>
> Dear reviewer knNF
>
> We appreciate the effort and time in reviewing the paper and providing comments.
> We carefully reviewed the comments and tried to respond to them.
> Since the revision period will end soon, we would like to know if there are further concerns or suggestions.
>
> Thank you!
>
> Best regards,
>  ㅤㅤㅤAuthors

---

### Meta-Review · Area_Chair_5yVx · 2023-12-10

**Metareview:**

This work introduces a denoising-/masked-autoencoder approach to SSL for ECG signals. The proposed method is grounded in considerations of the ECG domain, works well and is practical.

The major strength of this paper is to re-think SSL for challenging non-stationary time-series problems highlighting that SSL needs to be modality specific and ECG is a setting where labeled data can be expensive.

The exposition of the paper is clear, the method has been compared to many relevant baselines and outperforms them. Finally, the authors were able to address most reviewer concerns. Hence, the overall recommendation is for this paper to be published at ICLR.

**Justification For Why Not Higher Score:**

The method re-uses various known components and applies them in a clever way to ECG for SSL. While the work itself is important, it is not groundbreaking enough to warrant an oral presentation.

**Justification For Why Not Lower Score:**

All reviewers agreed, that the work is well-executed, well-grounded in related work and the method performs well. Hence, there is no reason to reject this paper.

---

### Decision · Program_Chairs · 2024-01-16

Accept (poster)